# BNP-Track: a framework for superresolved tracking

**Ioannis Sgouralis** [1,8], **Lance W. Q. Xu** [2,3,8], **Ameya P. Jalihal**[4], **Zeliha Kilic** [5], **Nils G. Walter** [6] **& Steve Pressé** [2,3,7] ✉

Superresolution tools, such as PALM and STORM, provide nanoscale localization accuracy by relying on rare photophysical events, limiting these methods to static samples. By contrast, here, we extend superresolution to dynamics without relying on photodynamics by simultaneously determining emitter numbers and their tracks (localization and linking) with the same localization accuracy per frame as widefield superresolution on immobilized emitters under similar imaging conditions (≈50 nm). We demonstrate our Bayesian nonparametric track (BNP-Track) framework on both in cellulo and synthetic data. BNP-Track develops a joint (posterior) distribution that learns and quantifies uncertainty over emitter numbers and their associated tracks propagated from shot noise, camera artifacts, pixelation, background and out-of-focus motion. In doing so, we integrate spatiotemporal information into our distribution, which is otherwise compromised by modularly determining emitter numbers and localizing and linking emitter positions across frames. For this reason, BNP-Track remains accurate in crowding regimens beyond those accessible to other single-particle tracking tools.

Characterizing macromolecular assembly kinetics[1], quantifying intracellular motility[2–4] or interrogating pairwise biomolecular interactions[5] requires accurate decoding of spatiotemporal processes at single-molecule scales, that is, high-nanometer spatial and rapid, often millisecond, temporal scales. These tasks ideally require superresolving positions of dynamic targets, typically fluorescently labeled molecules (light emitters), to tens of nanometer spatial resolution[6–9] and, when more than one target is involved, discriminating between signals from multiple targets simultaneously.

Assessments using fluorescence experiments at the required scales suffer from inherent limitations often arising from the diffraction limit of light (≈250 nm in the visible range for typical applications), below which conventional fluorescence techniques cannot resolve neighboring emitters. To overcome limitations of conventional tools and achieve superresolution, improvements have been achieved

through structured illumination[10], structured detection[11–14], photoresponse of fluorophore labels to excitation light[6–8,15,16] or combinations thereof[17–20].

Here, we focus on widefield superresolution microscopy (SRM), which typically relies on fluorophore photodynamics to achieve superresolution. SRM is regularly used both in vitro[21,22] and in cellulo[7,8,23–26]. Specific widefield SRM image acquisition protocols, such as STORM[15], PALM[8] and PAINT[21], through their associated image analyses, decode positions of light emitters separated by distances below the diffraction limit, often down to tens of nanometer resolution[8,15]. These widefield SRM protocols can be broken down into the following three conceptual steps: (1) specimen preparation, (2) imaging and (3) computational processing of the acquired images (frames). The success of step 3 is ensured by steps 1 and 2. In particular, in step 1, engineered fluorophores are selected, enabling the desired

[1]Department of Mathematics, University of Tennessee, Knoxville, TN, USA. [2]Center for Biological Physics, Arizona State University, Tempe, AZ, USA. [3]Department of Physics, Arizona State University, Tempe, AZ, USA. [4]Department of Cell Biology, Duke University, Durham, NC, USA. [5]Department of Structural Biology, St. Jude Children's Research Hospital, Memphis, TN, USA. [6]Single Molecule Analysis Group and Center for RNA Biomedicine, Department of Chemistry, University of Michigan, Ann Arbor, MI, USA. [7]School of Molecular Sciences, Arizona State University, Tempe, AZ, USA. [8]These authors contributed equally: Ioannis Sgouralis, Lance W. Q. Xu. ✉e-mail: spresse@asu.edu

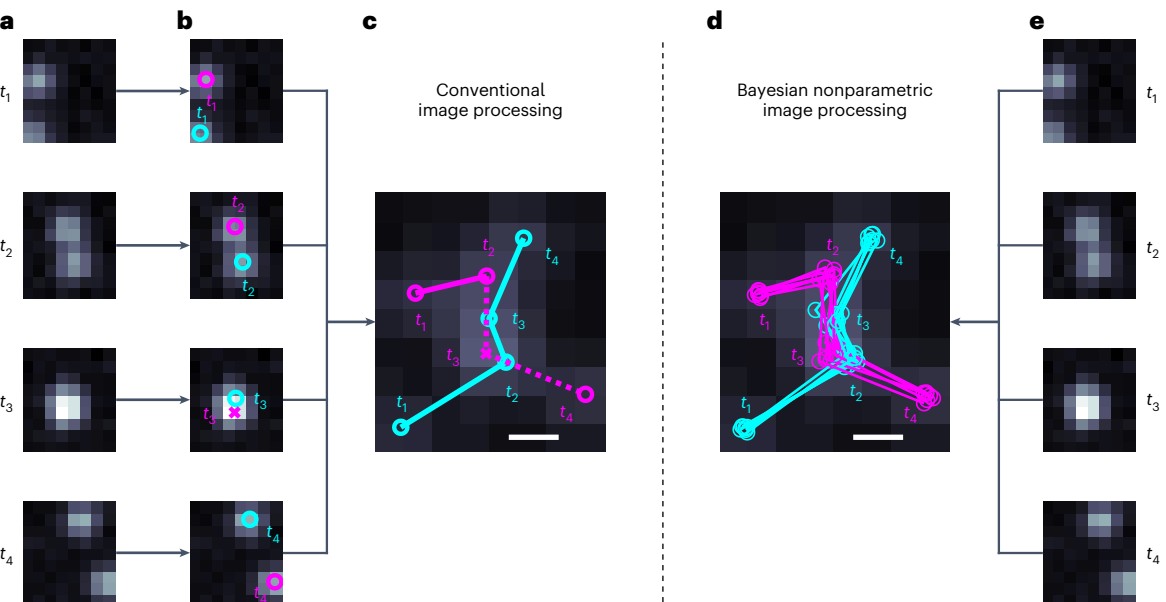

**Fig. 1 | Conceptual comparison between widely available tracking frameworks and BNP-Track. a,e**, Four frames from a dataset showing two emitters. **b**, Existing tracking approaches either completely or partially separate the task of first identifying and then localizing light emitters in the FOV of each frame independently. **c**, Conventional approaches then link emitter positions across frames. **d**, Our nonparametric approach (BNP-Track) simultaneously determines the number of emitters, localizes them and links their positions across frames. In **b**–**d**, circles denote correctly identified emitters, and crosses (×) denote missed emitters. In **c** and **d**, the scale bars indicate a distance equal to the nominal diffraction limit given by the Rayleigh diffraction limit of $0.61\lambda/NA$.

photodynamics, for example, photoswitching in STORM[15], photoactivation/photobleaching in PALM[8] or fluorophore binding/unbinding in PAINT[21]. Step 2 is then performed over extended periods, while rare photophysical (or binding–unbinding) events occur, and sufficient photons are collected to achieve superresolved localizations in step 3. For well-isolated bright spots, step 3 achieves superresolved localization[6,7,27] while accounting for effects such as light diffraction, resulting in spot sizes of roughly twice $0.61\lambda/NA$ (the Rayleigh diffraction limit), set by the emitter wavelength ($\lambda$), the microscope objective's numerical aperture (NA)[28], the camera and its photon shot noise and spot pixelization.

Here, we show that computation alone may overcome the reliance on the photophysics of step 1 and the long acquisition times of step 2, which not only largely limit widefield SRM to spatiotemporally fixed samples but also induce sample photodamage. For example, although a moving emitter's motion blur distributed over frames and pixels is typically a net disadvantage in the implementation of step 3, we conversely demonstrate that such a distribution of the photon budget in both space and time provides information that can be leveraged to superresolve emitter tracks, determine emitter numbers and help discriminate targets from their neighbors, even in the complete absence of photophysical processes (Fig. 1).

Although captured in more detail in the framework put forward in Methods and Supplementary Information, here, we briefly highlight how our tracking framework, Bayesian nonparametric Track (BNP-Track), fundamentally differs from conventional tracking tools that determine emitter numbers, localize emitters and link emitter locations in sequential (modular) steps. In the language of Bayesian statistics, resolving emitter tracks and emitter numbers amounts to constructing the probability distribution $\mathbb{P}($ links, locations, emitter numbers|data), which reads 'the joint posterior probability distribution of emitter numbers, locations and links given data'. The best set of emitter numbers and tracks are those globally maximizing this probability distribution. Without further approximation, this probability distribution can be decomposed as the following product:

$$
\begin{aligned}
&\mathbb{P}\,(\text{links, locations, emitter numbers}|\text{data})\\
&= \mathbb{P}\,(\text{links}|\text{locations, emitter numbers, data})\\
&\quad \times \mathbb{P}\,(\text{locations}|\text{emitter numbers, data})\\
&\quad \times \mathbb{P}\,(\text{emitter numbers}|\text{data})\,.
\end{aligned}
\tag{1}
$$

Single-particle tracking (SPT) tools performing emitter number determination, emitter localization and linking as separate steps (for example, see refs. [6,7,29–36] and many more reviewed therein) invariably approximate the joint distribution's maximization as a serial maximization of three terms. This process often involves additional approximations, such as using $\mathbb{P}($ links|locations) to approximate $\mathbb{P}($ links|locations, emitter numbers, data). Approximations such as these are acceptable for well-isolated and in-focus emitters. However, they have fundamentally limited our ability to superresolve emitters, especially as these move within light's diffraction limit. By contrast, BNP-Track avoids such approximations and leverages all sources of information to construct the joint posterior, yielding superresolved emitter tracks.

The overall input to BNP-Track includes both raw image sequences and known information on the imaging system, including the microscope optics and camera electronics, as further detailed in Methods. Using Bayesian nonparametrics, we estimate unknowns, including the number of emitters and their associated tracks. We demonstrate BNP-Track on experimental SPT data, detailing how the simultaneous determination of emitter numbers and tracks can be computationally achieved. We also benchmark BNP-Track's performance in Results against TrackMate[30], to which we confer some advantage because direct comparison is impossible as existing tools do not simultaneously learn emitter numbers and associated tracks.

## Results

To demonstrate our approach, we use BNP-Track first to analyze single mRNA molecules diffusing in live U-2 OS cells imaged under

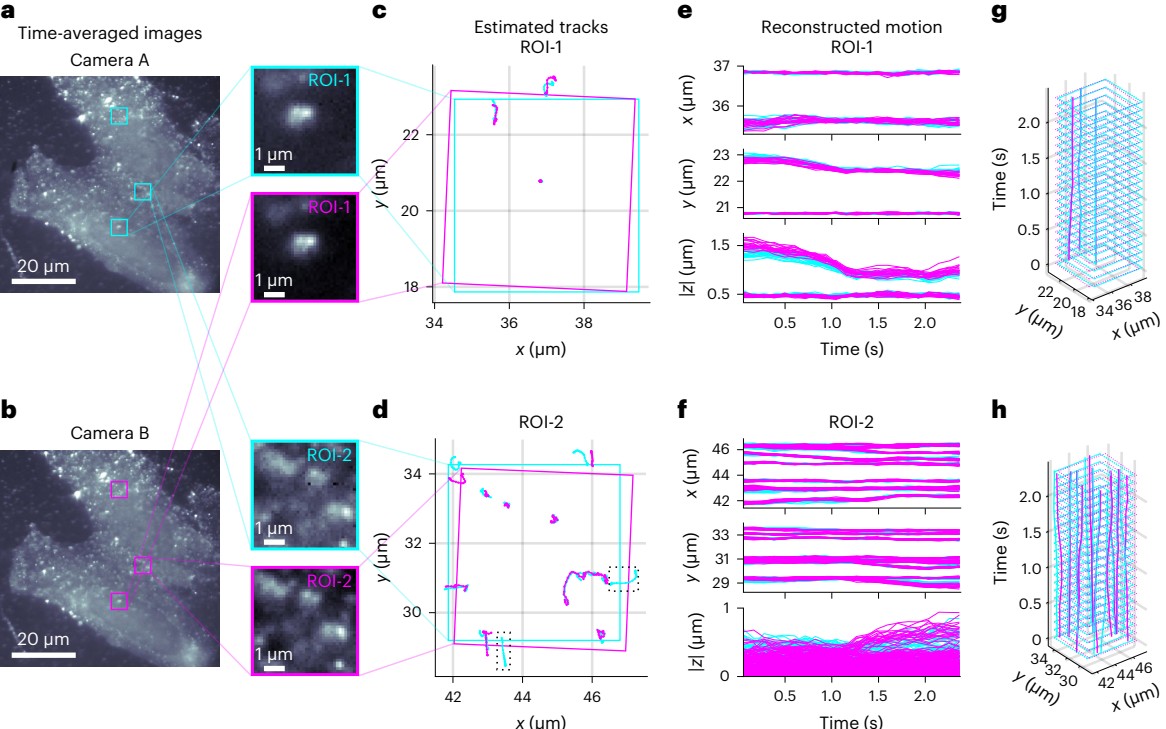

**Fig. 2 | Testing BNP-Track's performance on two 5-μm-wide regions of interest with different emitter densities based on an experimental dataset from fluorophore-labeled mRNA molecules diffusing in live U-2 OS cells onto a dual-camera microscope. a,b,** For convenience only, we show time averages of all 22 frames analyzed from cameras A (**a**) and B (**b**). The selected ROIs are boxed, and the zoomed-in images of the indicated ROIs are shown on the right. The remaining region, ROI-3, is only highlighted and is analyzed later in the text (Fig. 5). **c,d,** Estimated tracks within the selected ROIs from both cameras, with solid boxes indicating the corresponding ROIs after image registration. **e,f,** Reconstructed time courses for individual tracks from the selected ROIs. The dotted boxes in **d** highlight two emitter tracks only detected by camera A. **g,** Time course reconstruction by combining the top and middle of **e**. **h,** Time course reconstruction by combining the top and middle of **f**.

single-plane HILO illumination[37] on a fluorescence microscope as previously described[2,38,39] and use a beamsplitter to divide the single-color signal onto two cameras (Fig. 2). The dual-camera setup allows us to test for consistency of BNP-Track's emitter number and track determination across cameras. In subsequent tests, we use noise-overlaid synthetic data for which the ground truth is known[30,34] (Figs. 3 and 4 and Supplementary Fig. 2) and finally challenge BNP-Track with experimental data of crowded emitters (Fig. 5).

As emitters move in three dimensions (3D), it is possible, and indeed helpful in more accurate lateral localization, for BNP-Track to estimate emitter axial distance ($|z|$) from the in-focus plane from two-dimensional (2D) images by modeling the dependence of the width of the emitter's point spread function (PSF) on $|z|$[40]. For this reason, although the axial distance from the in-focus plane is always less accurately determined than the lateral positions, we nonetheless report BNP-Track's axial estimates for experimental data in Figs. 2 and 4.

Before showing the results, we note an important feature of Bayesian inference. Developed within the Bayesian paradigm[41–43], BNP-Track goes beyond providing mere point estimates for unknown variables like the number of emitters and their corresponding tracks. It offers posterior probability distributions for these quantities, from which 95% credible intervals (CIs) can be computed. As we cannot easily visualize the output of multidimensional distributions over all candidate emitter numbers and associated tracks, we often report estimates for emitters that coincide with the number of emitters maximizing the posterior, termed maximum a posteriori (MAP) point estimates[44]. Having determined the MAP number of emitters, we then collect their associated tracks in figures, such as in Figs. 2 and 5.

## BNP-Track superresolves sparse emitter tracks in cellulo

Because no direct ground truth is available for tracks from experimental SPT data, we use two cameras behind a beamsplitter to assess the success of BNP-Track. Using image registration (to correct for camera misalignment), we independently process two datasets for subsequent comparison and error estimation, knowing that, in principle, both cameras should have the same tracks (our ground truth). However, the noise realizations on both cameras are different, and emitters may move closer to each other than the nominal diffraction limit of 231 nm. To estimate tracking error quantitatively, based on Chenouard et al.[34], we define a tracking error metric; see Eqs. (2) to (4) in Methods for details.

In Fig. 2a,b, we show, for illustrative purposes alone, time averages of a sequence of 22 successive frames spanning ≈2.5 s of real time in both detection channels, reflecting the complexity of the data to which BNP-Track applies. In data processing, we analyze the underlying frames without averaging. All raw data are provided in Supplementary Data 1–4.

From these frames, we track well-separated or dilute emitters (that is, whose PSFs are always well separated in space) in a 5-μm-wide square region of interest (ROI), named ROI-1. Fig. 2a also zooms in on ROI-1. As these are real experimental image stacks, there is no reason to assume a priori that the motion model is normally diffusive. The BNP-Track-derived track estimates are shown in Fig. 2c, while in Fig. 2e, all samples drawn from the posterior distribution are superposed. Due to fundamental optical limitations, we cannot determine whether the emitter's axial position lies above or below the in-focus plane, so we only report the absolute value of the emitters' axial position. As evident from Fig. 2c, BNP-Track successfully identifies and localizes the same tracks within the selected ROI in the two parallel camera

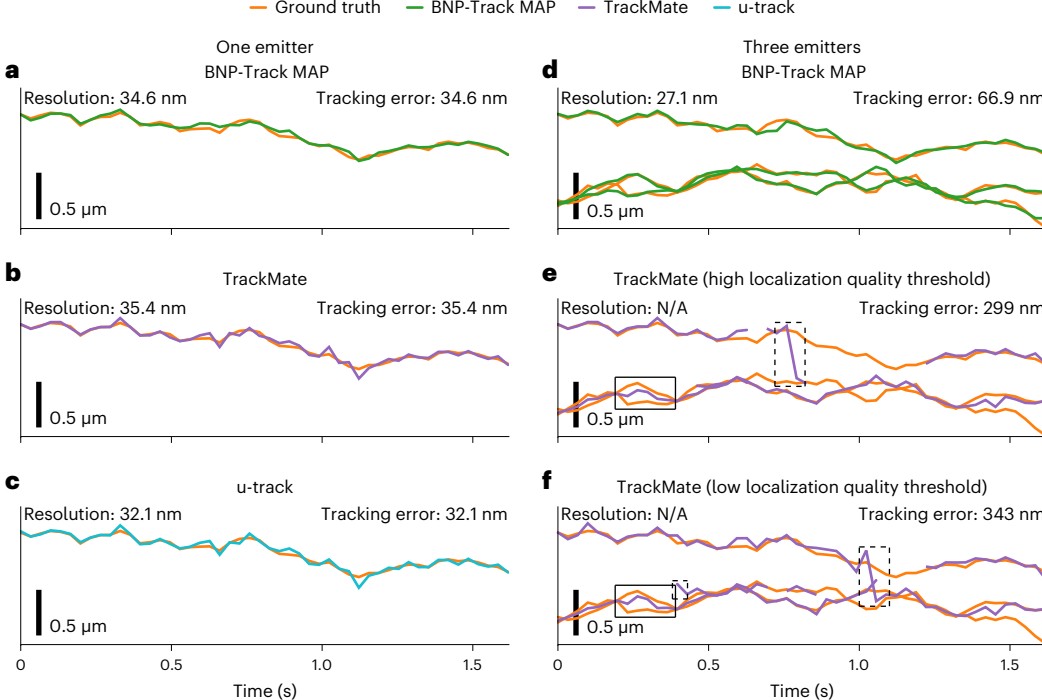

**Fig. 3 | A comparison of tracking performance among BNP-Track, TrackMate and u-track using two synthetic datasets with one emitter and three emitters in the *y* coordinate.** See Extended Data Fig. 2 for the same figure but with the *x* coordinate. **a**, BNP-Track's MAP estimate compared to the one-emitter ground truth. **b**, TrackMate's estimate compared to the one-emitter ground truth. Sets A and B are equivalent in this case. **c**, u-track's estimate compared to the one-emitter ground truth. **d**, BNP-Track MAP estimates compared to the three-emitter ground truth. **e**,**f**, TrackMate estimates compared to the three-emitter ground truth with high localization quality threshold (**e**) and low localization quality threshold (**f**). In **d**–**f**, the top track of ground truth is the same as the ground truth in **a**–**c**. The boxed regions in **e** and **f** highlight areas where TrackMate performs relatively poorly. See Supplementary Data 5 and 6. Localization resolution is not applicable (N/A) to TrackMate estimates due to missing track segments.

datasets of ROI-1 despite different background and noise realizations on each camera. Of note, the two square ROI-1s are rotated relative to one another based on image registration in postprocessing. Fig. 2e shows tracks of all well-separated emitters identified within this field of view (FOV). We note that despite that the center of an emitter's PSF near the top of ROI-1 in Fig. 2c lies outside ROI-1 for both cameras in all 22 frames, it is surprisingly independently picked up in the analysis of the data from both cameras. However, we exclude this unique track outside the FOV from Fig. 2e and further analysis.

Using the metric defined above in Eq. (4) for Fig. 2e, we report a tracking error of about 37 nm in the lateral direction, consistent with the prior superresolution values for immobilized targets[6–8]. As BNP-Track provides estimates of the lateral as well as the magnitude of the axial emitter position (see details in Methods), we can also assess the full 3D tracking error. This results in a ≈48-nm 3D tracking error. Having shown that we can track emitters in a dilute regimen similar to ROI-1 of Fig. 2c,e,g, we next analyze a more challenging ROI, ROI-2, where emitter PSFs now occasionally overlap.

As before, for illustrative purposes only, in Fig. 2b we show time averages of a sequence of 22 successive frames spanning ≈2.5 s of real time. Fig. 2d,f,h reflects the same information as described for ROI-1 but for ROI-2. As before, BNP-Track tracks emitters even as these diffuse away from a camera's FOV. Using Eq. (4), we find that for ROI-2, the tracking error is slightly higher at 64 nm in the lateral direction, remaining below the nominal diffraction limit of 231 nm. Additionally, the tracking error in 3D now grows to 159 nm, resulting in a tracking error of about 80 nm.

BNP-Track also estimates other dynamical quantities, including the background photon flux (photons per unit area per unit time), emitter brightness (photons per unit time), (effective) diffusion coefficient and number of emitters. Estimates for these quantities are

summarized in Extended Data Fig. 1. From Extended Data Fig. 1b,c, the system's background flux and emitter brightness vary substantially over time. Despite the agreement between tracks deduced from both cameras below light's diffraction limit in ROI-2, discrepancies in some quantities (such as the diffusion coefficient in Extended Data Fig. 1d) highlight the sensitivity of these quantities to small track differences below light's diffraction limit. Similarly, small discrepancies in the emitter brightness estimates (Extended Data Fig. 1c) may be induced by minute dissimilarities in the optical path leading to each camera.

Finally, the number of emitters detected in the two cameras differs (Extended Data Fig. 1), with the additional tracks detected by camera A highlighted by dotted boxes in Fig. 2d. This is unsurprising for three reasons. First, the two cameras have slightly different FOVs. Second, as highlighted by the dotted boxes in Fig. 2d, a notable portion of the two extra tracks lies outside the FOV of either camera and thus are challenging to detect under any circumstance. Third, as out-of-focus emitters can mathematically model background noise and because two cameras draw slightly different conclusions on background photon emission rates and emitter brightnesses (Extended Data Fig. 1b,c), this may also naturally lead to slightly different estimates of the number of emitters, especially those out of focus or beyond the FOV. BNP-Track detects in-focus emitters and uses what it learns from in-focus emitters to extrapolate outside the FOV or in-focus plane. In such regions, the number of photons that BNP-Track uses to draw inferences on tracks is naturally limited.

## Benchmarking BNP-Track with synthetic data

Next, we validate BNP-Track by using synthetically generated data where ground truth tracks are known. To ensure realistic data, we adopt the procedure outlined in Methods and Supplementary Information for data generation. The parameters (NA, pixel size, frame

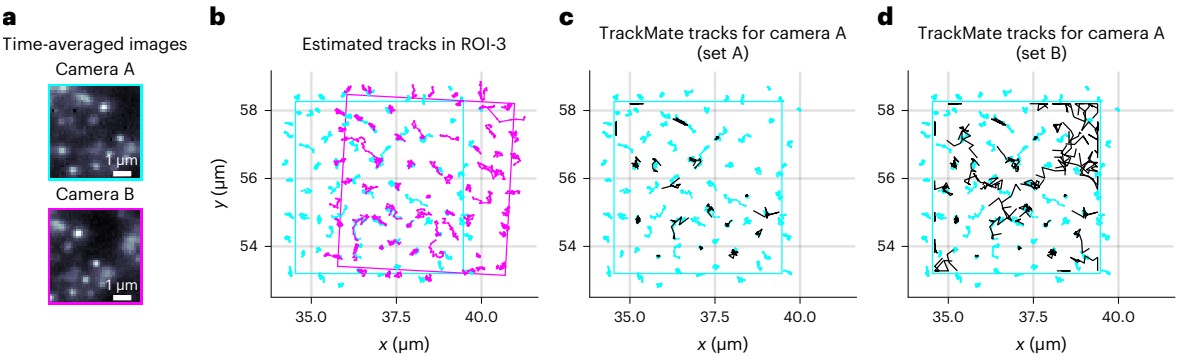

**Fig. 4 | Benchmarks of BNP-Track regarding the mean displacement between simulated tracks.** TrackMate tracks (set B) are also provided for comparison. **a**, Time-averaged image for the synthetic scenario (Supplementary Data 18) where the mean displacement between tracks is 724 nm. **b**, BNP-Track's reconstructed tracks for all coordinates. The reconstructed tracks are in low-opacity cyan, and the ground truths are magenta. Lateral tracking errors (see Eq. (3)) are listed at the top right. **c**, TrackMate's estimated tracks (low-opacity green) of the same datasets with the ground truths (magenta).

Lateral tracking errors are listed at the top right. Also, no axial ($|z|$) result is plotted in **c** as TrackMate does not provide axial tracks for 2D images. **d**–**f**, The same layout as **a**–**c** but with a mean displacement of 579 nm (Supplementary Data 19). **g**–**i**, The same layout as **a**–**c** but with a mean displacement of 434 nm (Supplementary Data 20). **j**–**l**, The same layout as **a**–**c** but with a mean displacement of 289 nm (Supplementary Data 21). **m**–**o**, The same layout as **a**–**c** but with a mean displacement of 145 nm (Supplementary Data 22). **p**–**r**, The same layout as **a**–**c** but with a mean displacement of 0 nm (Supplementary Data 23).

**Fig. 5 | BNP-Track's performance in increasingly crowded environments.**
**a**, Time-averaged images over 22 frames from both cameras show many diffraction-limited emitters within the ROI analyzed here (ROI-3). **b**, BNP-Track's tracking results after registering FOVs. **c**, Comparison between the BNP-Track result and the TrackMate result with a high localization quality threshold

(set A) for the data from camera A. TrackMate tracks are in black; see Tinevez et al.[30] for the definition of quality. **d**, A comparison between the BNP-Track result and the TrackMate result with a low localization quality threshold (set B) for the data from camera A. TrackMate tracks are in black.

rate, diffusion coefficient, emitter brightness and background photon flux) used in generating synthetic data are identical or similar to those used in the earlier experiments. All parameter values are specified in

Supplementary Table 1. Using simulated data with knowledge of the ground truth tracks, we evaluate BNP-Track's performance in two ways. First, we compare BNP-Track's tracking accuracy to an established

SPT tool built on a leading tracking tool[34], TrackMate[30]. Second, we test BNP-Track's robustness across different parameter regimens and motion models (beyond normal diffusion).

Comparing BNP-Track to other SPT methods fairly and directly poses challenges. For instance, no other existing method simultaneously estimates emitter numbers alongside their associated tracks (alongside diffusion coefficients, time-dependent emitter brightnesses and time-dependent background photon fluxes). We simulated data with emitter brightness and background photon flux constant over time to address this. In the analyses conducted by BNP-Track, we fix the emitter brightness and no longer estimate its value, although diffusion coefficients and the constant background flux are still inferred. By contrast, the ground truth values for emitter brightness, background photon flux and diffusion coefficient are supplied to the conventional SPT tools. This deliberate approach gives a substantial advantage to these tools.

Additionally, considering that the dimension of the SPT problem (the number of estimated variables) can easily exceed hundreds (compare Eq. (1)), tuning the parameters of any participating SPT tool to optimize their tracking performance becomes impractical within finite time. Consequently, we allow 1 day to optimize tracking performance for BNP-Track, u-track and TrackMate. This limitation rules out options including writing customized code to optimize the tracking performance of conventional SPT tools.

Although the numerical value of the posterior distribution is an important performance metric for BNP-Track, no single metric exists to assess the performance of conventional SPT tools[45]. Without a quantitative numerical criterion such as a posterior value, we rely on preselected metrics, for example, tracks with minimal spurious detections or the fewest missed links (termed false negatives). Also, although it is generally preferable to have tracks with no false positives (spurious detections or tracks) and no false negatives (missed detections or tracks), competing methods often struggle to achieve both simultaneously. This difficulty arises because most methods cannot set frame-specific thresholds. Consequently, when false negatives and false positives cannot be reduced to 0 simultaneously for a conventional SPT tool, we output two sets of tracks for this SPT tool from the data. Set A, with ground truth tracks presented, prioritizes minimizing false positives during the localization (spot detection) phase and subsequently minimizes false negatives during the linking phase. By contrast, set B, again with ground truth tracks presented, focuses on reducing false negatives during the localization phase and then addresses false positives during the linking phase. Finally, if the resulting tracks consist of separate segments belonging to a single ground truth track, we manually fuse them, providing another advantage to competing tools.

Consequently, to summarize, throughout this study, we give a critical advantage to existing tools that we compare to BNP-Track by (1) manually tuning the parameters of these tools to have them best match the ground truth emitter numbers, locations and links and (2) asking BNP-Track to estimate parameters (diffusion coefficient and background photon flux) from the data while the ground truth values for these same parameters are used to tune competing methods to optimize their performance. See Methods for exactly how the aforementioned parameters are provided to existing methods and how track segments are fused. As we will show, although competing methods have substantial advantages, BNP-Track still exceeds the resolution of existing tools and yields reduced error rates (percentage of wrong links).

To quantitatively compare SPT tools to BNP-Track, we continue using the pairing distance and tracking error metrics previously used (Eqs. (2) to (4)). In addition, we use a finite gate value (see $\epsilon$ in Eq. (2)) of five pixels (presented in Methods). This gate value allows us to benchmark SPT tools that otherwise face challenges in localizing emitters within specific frames (that is, have missing segments) using a defined threshold. Using reasonably different gate values does not alter the subsequent discussion (see Supplementary Tables 2 and 3).

Nevertheless, pairing distance and tracking error metrics do not fully reflect track quality because linking error (mislink) penalties are included in the assessment. Including mislinks can be problematic when two or more emitters largely overlap in a frame, with emitters losing their identities and becoming indistinguishable. In such cases, penalizing mislinks is irrelevant to assessing overall track quality.

Therefore, to better capture localization accuracy, we introduce another metric called localization resolution, or simply resolution, that does not consider mislinks. Instead of pairing tracks, localization resolution independently pairs emitter positions in each frame without using a gate value. Consequently, localization resolution is undefined if the compared tracks do not have the same number of emitters in any given frame. In addition, if there are no mislinks, false positives or false negatives, tracking error and localization resolution are equivalent.

## Comparison with TrackMate
The study begins with tracking a single emitter, where both BNP-Track (Fig. 3a), TrackMate (Fig. 3b) and u-track (Fig. 3c) accurately follow the emitter throughout a video sequence, achieving resolutions around 34.6 nm, 35.4 nm and 32.1 nm, respectively, demonstrating typical performance in straightforward scenarios. As the complexity increases to a three-emitter setup (see Supplementary Fig. 1g and Supplementary Data 6), BNP-Track outperforms TrackMate under most criteria (Fig. 3d and Supplementary Table 3), particularly in situations with overlapping PSFs, maintaining a resolution of 27.1 nm. TrackMate, however, struggles with issues like the misinterpretation of diffraction-limited emitters and spurious detections (Fig. 3e,f), which severely impairs its ability to resolve emitters, leading to notable tracking errors and diffusion coefficient overestimates. Detailed data and further analyses, including comparative metrics and the impact of tracking errors, are available in the Supplementary Information.

## Robustness tests
We present in the Supplementary Information robustness results for BNP-Track by considering data drawn from different motion models and varying diffusion coefficients for normal diffusion in addition to varying emitter brightnesses, background photon fluxes and emitter numbers. As an example, in Fig. 4, we test how closely two emitters can come together while retaining the ability of BNP-Track to enumerate the number of emitters and track them.

## BNP-Track's performance in increasingly crowded environments
So far, we have evaluated the performance of BNP-Track on two distinct ROIs from an experimental dataset and computed its resolution using synthetic data. To further test the limits of BNP-Track, we analyzed a densely packed ROI, ROI-3 (Supplementary Data 30 and 31), selected from the same data set (Fig. 2a).

In Fig. 5a, similar to Fig. 2a,b, for illustrative purposes alone, we show time-averaged images from both cameras. These images reveal that ROI-3 contains tens of closely positioned and out-of-focus emitters. Furthermore, within ROI-3, cameras A and B observe slightly different FOVs, offset by approximately 1.5 µm and rotated by 5°. To provide an assessment of BNP-Track's performance compared to that of TrackMate, we selected emitters whose z positions are within 150 nm of the in-focus plane in the overlapping region and calculated the pairing distance between tracks. The results show a tracking error of approximately 136.4 nm, which corresponds to a tracking error of 68.2 nm compared to the ground truth. These results are consistent with the performance for ROI-2 in Fig. 2.

As illustrated in Fig. 5a,b, ROI-3 presents a challenge in estimating the number of emitters due to crowding and overlapping PSFs as well as a larger number of out-of-focus emitters. These features pose severe challenges to conventional tracking tools that rely on manually setting thresholds to fix the number of emitters, especially dim

or out-of-focus ones. To demonstrate this point, we used TrackMate again to analyze ROI-3. The results for camera A are illustrated in Fig. 5c,d. Specifically, for Fig. 5c, we set a high localization quality threshold (set A) relative to the nominal diffraction limit of 231 nm at 5 pixels (665 nm) to minimize spurious detection, resulting in a total of 18 tracks. Each of these tracks can be paired with one of BNP-Track's 78 emitter tracks using the Tracking Performance Measures tool[34] in Icy[46], with the maximum pairing distance set at 2 pixels (266 nm). Despite the high localization threshold, TrackMate produces notably fewer emitter tracks than BNP-Track due to out-of-focus dim emitters and difficulties arising from overlapping PSFs. By contrast, we lower the quality threshold in TrackMate to 0 to detect more emitters (set B), thus increasing the total number of TrackMate tracks to 64 (Fig. 5d). Forty-one tracks can be paired with a subset of BNP-Track's emitter tracks using the same pairing distance threshold. However, 23 spurious tracks are contaminating further analysis. Similar results are also obtained for data from camera B (Extended Data Fig. 3a,b). Furthermore, in Extended Data Fig. 3c,d, we show that, given the same image registration as used in Fig. 5b, TrackMate does not produce matching tracks for both cameras, underscoring why emitter numbers must be simultaneously learned while tracking rather than precalibrating emitter numbers.

## Discussion

We present an image processing framework, BNP-Track, superresolving emitters in cellulo without leveraging fluorophore photodynamics. Our framework analyzes continuous image measurements from diffraction-limited light emitters throughout image acquisition. BNP-Track extends the scope of widefield SRM by exploiting spatiotemporal information encoded across all frames and pixels. Additionally, BNP-Track unifies many existing approaches to localization microscopy and SPT and extends beyond them by simultaneously and self-consistently estimating emitter numbers.

By operating in three interlaced stages (preparation, imaging and processing), existing approaches to widefield SRM estimate locations of individual static emitters with a generally accepted resolution of ≈50 nm or less[6–8]. Such resolution for widefield applications is substantially improved relative to conventional microscopy's diffraction limit of ≈250 nm. Although our processing framework cannot lift the limitations imposed by optics nor eliminate the degradation induced by noise, we show that our framework can substantially extend our ability to estimate emitter numbers and tracks from existing images with uncertainty both for more straightforward in-focus cases where emitters are well separated and more challenging cases where emitters are crowded, move out of focus and appear partly out of the FOV. In particular, because BNP-Track provides full distributions over unknowns, it readily computes error bars (often termed CIs within a Bayesian setting) associated with emitter numbers warranted by images (for example, Extended Data Fig. 1 and Supplementary Fig. 1), localization events for both isolated emitters and emitters closer than light's diffraction limit and other parameters including diffusion coefficients.

Many tracking scenarios challenge all tracking tools, including out-of-focus motion, crowded environments, inhomogeneous illumination, optical aberrations, hot pixels or detector saturation. Although quantifying when BNP-Track fails depends on the specifics of any given circumstance, BNP-Track leverages broad spatiotemporal information typically eliminated by separating the tracking task in modular steps by traditional tools as highlighted earlier. BNP-Track leverages all information available by modeling the entire process from emitter motion to detector output simultaneously and self-consistently, thus maximizing the amount of information extracted from individual frames. As such, when BNP-Track fails to track for a particular system setting, conventional tracking methods typically fail earlier (for example, as shown in Fig. 4), indicating the need for an alternative experimental protocol.

The analysis of FOVs like those shown in Results (5 μm × 5 μm or about 1,500 pixels and 22 frames) requires about 300 min of computational time on an average laptop (Apple MacBook 2020 with macOS Ventura). The computational cost scales linearly with frame number, total pixel number and total emitter numbers. Larger-scale applications are within the existing computational capacity, although additional algorithmic improvements and computational optimization are possible. For instance, as part of future technological development, parallelism can be used in Eq. (6), the most costly part of our algorithm, given that the average photon numbers incident on each pixel are independent of the others. This approach results in a theoretical speed-up factor equal to the smaller of the two numbers: the total number of pixels and the number of processor cores. What is more, as single-photon detector arrays become available, it may be possible to imagine generalizing our emission model to consider binary (detection or nondetection) or other few-bit emission models extending beyond the continuum emission model invoked herein, thereby generalizing BNP-Track to faster diffusion processes.

Despite BNP-Track's higher computational cost, we argue the following. First, as demonstrated in Figs. 3 to 5, cheaper conventional tracking methods not only fail to surpass the diffraction limit but also do not learn emitter numbers. However, learning emitter numbers is especially critical in correctly linking emitter locations across frames, especially in crowded environments. Second, BNP-Track's execution time is primarily computational wall time, as BNP-Track is unsupervised and largely free from manual tuning. This stands in contrast to methods such as TrackMate used in the generation of Figs. 3 to 5, which require manual tuning and thresholding for proper execution and, even so, remain diffraction limited.

As BNP-Track is a framework, it can be adapted to accommodate specialized illumination modalities including TIRF[47] and light-sheet[48] or even multicolor imaging. Indeed, microscopy modalities collecting data across axial planes may help discriminate between background and out-of-focus emitters that are currently difficult to distinguish. For example, BNP-Track attributes variations in the appearance of spots (changes in shape, size and emission intensity) to (1) emitters moving in and out of the in-focus plane, (2) overlapping emitters and (3) motion blur. Along these same lines, in Methods, we made common modeling choices and used typical experimental parameters. For example, we used an EMCCD camera model and assumed a Gaussian PSF. Other choices can be made by simply changing the mathematical form of the camera model or the PSF, provided that these assume known precalibrated forms. None of these changes break BNP-Track's conceptual framework.

Similarly, while BNP-Track uses a Brownian motion model, one may wonder about BNP-Track's performance when emitters evolve according to alternative motion models. A preliminary answer to this question lies in the Results, where BNP-Track yields accurate tracking results consistent across two cameras for an experimental dataset with an unknown emitter motion model despite assuming normal diffusion (Brownian) motion. Moreover, further simulations shown in Supplementary Figs. 3 and 7 to 9 illustrate how BNP-Track applies and maintains its performance in cases where emitter motion is dictated by motion models extending beyond Brownian motion, highlighting the dominant contribution of the detector and photon emission model in tracking over the details of the motion model. This motivates our thinking of Brownian motion as justifying the use of Gaussian transition probabilities, following from the central limit theorem, between locations across frames.

Perhaps more fundamentally, these results imply that the amount of diffraction-limited tracking data analyzed may be insufficient to infer motion models, given that tracks learned by BNP-Track remain accurate even when the underlying motion model differs from normal diffusion. Either way, if we believe that a specific motion model is warranted and not accommodated by Gaussian transition probabilities,

we may incorporate this change into our framework (Supplementary Note 6.3). The current questions raised by the insensitivity of the track determination to the motion model do raise questions as to how sensitive tracking is to boundary conditions of cells and obstacles encountered within cells, which we have yet to explore.

Building on the results and discussions presented thus far, we posit that BNP-Track applies to a broad spectrum of particle tracking scenarios. Its utility becomes especially pronounced in systems where traditional modular localization and linking faces challenges, specifically in scenarios featuring relatively dim emitters, intense background noise, fast diffusing emitters or a high local emitter number density. For systems containing consistently well-separated bright in-focus emitters within a large FOV, conventional tracking approaches may be preferred. Yet, it is difficult to control a fixed separation between emitters, even under low crowding conditions.

Postprocessing tools are frequently used to extract useful information from single-particle tracks, such as diffusion coefficients and diffusive states. These tools range from simple approaches, such as MSD, to more complex methods, such as Spot-On[49] and SMAUG[50]. Because our framework produces tracks, these tracks can be analyzed by these tools, and our ability to make full distributions over tracks may also help estimate errors over postprocessed parameters. It is also conceivable that our output could be used as a training set for neural networks[51] or be used to make predictions of molecular tracks in dense environments[52], such as in Fig. 5, previously considered outside the scope of existing tools.

The framework that we present here is a proof-of-principle demonstration that computation feasibly achieves superresolution of evolving targets by avoiding the existing tracking paradigm's modular structure and limiting tracking to dilute and in-focus samples.

## Online content

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

## Methods

### Tracking error metrics

The agreement between tracks across both cameras within their FOV can be quantified according to a pairing distance metric[34]. Briefly, the distance between two tracks is defined using the gated Euclidean distance given by

$$d_\epsilon(\psi_{1:N}, \phi_{1:N}) = \sum_{n=1}^{N} \min(\| \psi_n - \phi_n \|, \epsilon). \qquad (2)$$

Here, $\epsilon$ is called the gate value (representing the maximum distance for two detections to be paired), $N$ is the total number of frames, and $\|\psi_n - \phi_n\|$ denotes the Euclidean distance between the emitter positions $\psi_n$ and $\phi_n$ at time $t_n$ in both cameras. If a track fails to localize any emitter in a particular frame, the distance at that frame is considered to be $\epsilon$ (numerically defined shortly). When comparing BNP-Track's tracking results for both cameras in the dual-camera setup, we use an infinite value for $\epsilon$. This is because, by design, BNP-Track only outputs tracks with no missing segments, as emitters are assumed to originate from outside of the FOV or out of focus.

On the other hand, when comparing the track estimates from SPT tools to known ground truth tracks, the gate value should be configured to ensure that detections deemed 'well separated' or those exceeding twice the nominal diffraction limit are never paired. In the synthetic datasets, the nominal diffraction limit is approximately 280 nm, derived using parameters outlined in Supplementary Table 1. Consequently, we selected a gate value of 5 pixels, equivalent to approximately 665 nm. Despite this argument, using reasonably different gate values does not alter the subsequent discussion (refer to Supplementary Tables 2 and 3).

Based on Eq. (2), the pairing distance between two sets of tracks, denoted $d_\epsilon\left(\psi_{1:N}^{1:M_\psi}, \phi_{1:N}^{1:M_\phi}\right)$, is defined as the minimum total gated Euclidean distance among all possible track pairings between the sets. In this context, $M_\psi$ and $M_\phi$ represent the number of tracks in each set. For a comprehensive understanding of the methodology used to determine this minimal distance, see Chenouard et al.[34].

Moreover, to enable a more straightforward comparison with the diffraction limit, we introduce the concept of tracking error, which is defined as

$$\text{tracking error}\left(\psi_{1:N}^{1:M}, \phi_{1:N}^{1:M_{\text{ref}}}\right) = \frac{d_\epsilon\left(\psi_{1:N}^{1:M}, \phi_{1:N}^{1:M_{\text{ref}}}\right)}{N \times M_{\text{ref}}}. \qquad (3)$$

Here, the notation is consistent with that of Eq. (2), and $\phi_{1:N}^{1:M_{\text{ref}}}$ is the reference track set, which is the ground truth track set when available.

For the dual-camera experimental dataset, given the absence of ground truth, the tracking error is calculated slightly differently,

$$\text{tracking error}\left(\psi_{1:N}^{1:M,\text{camera 1}}, \psi_{1:N}^{1:M,\text{camera 2}}\right) = \frac{d_\epsilon\left(\psi_{1:N}^{1:M,\text{camera 1}}, \psi_{1:N}^{1:M,\text{camera 2}}\right)}{2 \times N \times M}. \qquad (4)$$

Here, the extra factor of two appears in the denominator as the tracking error now sums the distances from both tracks, and thus the error per track is half. In practice, we have found that both track sets share the same number of tracks, $M$, because BNP-Track reports the same number of emitters in the shared FOV between the two cameras in the experimental datasets discussed below. Otherwise, if the detectors are very different and the number of tracks detected is not the same, then by convention, $M$ can be understood as the mean.

### Image processing

As we demonstrate in Results, our analysis goal is to determine the probability distribution termed the posterior, $p(\theta|w_{1:N}^{1:P})$. In this distribution, we use $\theta$ to gather the unknown quantities of interest, for instance,

emitter tracks and photon emission rates, and $w_{1:N}^{1:P}$ to collect the data under processing, for example, timelapse images. Below, we present how this distribution is derived and its underlying assumptions.

We first present a detailed formulation of the physical processes in forming the acquired images necessary in the above quantitative analysis to facilitate the presentation. This formulation captures microscope optics and camera electronics and can be modified to accommodate more specialized imaging setups. In this formulation, the unknowns of interest are encoded by parameters. Next, we present the mathematical tools needed to estimate values for the unknown parameters. That is, we address the core challenge in SRM arising from the unknown number of emitters and their associated tracks. To overcome the challenge of estimating emitter numbers, we apply Bayesian nonparametrics. Our approach differs from the likelihood-based approaches currently used in localization microscopy, allowing us to relax the SRM photodynamical requirements.

As several of the notions in our description are stochastic (for example, parameters with unknown values and random emitter dynamics), we use probabilistic descriptions. Although our notation is standard for the statistical community, we provide an introduction more appropriate for a broader audience in Supplementary Information.

### Model description

Our starting point consists of image measurements obtained in an SRM experiment denoted by $w_n^p$, where subscripts $n = 1,\ldots,N$ indicate the exposures, and superscripts $p = 1,\ldots,P$ indicate pixels. For example, $w_2^3$ denotes the raw image value, typically reported in analog-to-digital units or counts and stored in TIFF format, measured in pixel 3 during the second exposure. Similarly, $w_2^{1:P}$ denotes every image value (that is, entire frame) measured during the second exposure. Because the image values are related to the specimen under imaging, we aim to develop a mathematical model encoding the physical processes that relate the system imaged with the acquired measurements.

**Noise.** The recorded images mix electronic signals that depend only stochastically on an average amount of incident photons[53–56]. For commercially available cameras, the overall relationship, from incident photons to recorded images, is linear and contaminated with multiplicative noise that results from shot noise, amplification and readout. Our formulation below applies to image data acquired with EMCCD-type cameras, as commonly used in superresolution imaging[6,55]. However, the expression below can be modified to accommodate other detector architectures. Here, in our formulation,

$$w_n^p \,|\, u_n^p \sim \text{Normal}\left(\mu + \xi u_n^p, v + f\xi^2 u_n^p\right), \qquad (5)$$

where $u_n^p$ is the average number of photons incident on pixel $p$ during exposure $n$. The parameter $f$ is a camera-dependent excess noise factor, and $\xi$ is the overall gain that combines the effects of quantum efficiency, preamplification, amplification and quantization. The values of $\mu, v, \xi$ and $f$ are specific to the camera that acquires the images of interest, and their values can be calibrated as described in Supplementary Information.

**Pixelization.** As shot noise is already captured, $u_n^p$ depends deterministically on the underlying photon flux

$$u_n^p = \int_{t_n^{\min}}^{t_n^{\max}} dt \iint_{x_{\min}^p, y_{\min}^p}^{x_{\max}^p, y_{\max}^p} dx\, dy\, U(x, y, t), \qquad (6)$$

where $t_n^{\min}$, $t_n^{\max}$ mark the integration time of the $n$th exposure; $x_{\min}^p, x_{\max}^p, y_{\min}^p, y_{\max}^p$ mark the region monitored by pixel $p$; and $U(x, y, t)$ is the photon flux at position $x, y$ at time $t$. We detail our spatiotemporal

frames of reference in Supplementary Information.

**Optics.** We model $U(x, y, t)$ as consisting of background $U_{back}(x, y, t)$ and fluorophore photon contributions (that is, flux) from every imaged light emitter $U_{fluor}^m(x, y, t)$. These are additive,

$$U(x, y, t) = U_{back}(x, y, t) + \sum_{m=1}^{B} U_{fluor}^m(x, y, t). \tag{7}$$

Specifically, for the latter, we consider a total of $B$ emitters that we label with $m = 1, \ldots, B$. Each of our emitters is characterized by a position $X^m(t)$, $Y^m(t)$, $Z^m(t)$, all of which may change through time. Here, we use uppercase letters $X$, $Y$ and $Z$ for random variables and lowercase letters $x$, $y$ and $z$ for general variables (realizations of the corresponding random variables). Because the total number $B$ of imaged emitters is a critical unknown quantity, in the next section, we describe how we modify the flux $U(x, y, t)$ to allow for a variable number of emitters. In Supplementary Information, we describe how this flux is related to $X^m(t)$, $Y^m(t)$, $Z^m(t)$.

## Model inference
The quantities that we wish to estimate, for example, the positions $X^{1:B}(t)$, $Y^{1:B}(t)$, $Z^{1:B}(t)$, are unknown variables in the preceding formulation. The total number of such variables depends on the number of imaged emitters $B$, which in SRM remains unknown, thus prohibiting the processing of the images under flux $U(x, y, t)$. Because $B$ has such a subtle effect, we modify our formulation to make it compatible with the nonparametric paradigm of data analysis, allowing for processing under an unspecified number of variables[41,42,57–59].

In particular, following the nonparametric latent feature paradigm[41,42], we introduce indicator parameters $b^m$ that adopt only values 0 or 1 and recast $U(x, y, t)$ in the form

$$U(x, y, t) = U_{back}(x, y, t) + \sum_{m=1}^{M} b^m U_{fluor}^m(x, y, t). \tag{8}$$

Specifically, with the introduction of indicators, we increase the number of emitters represented in our model from $B$ to a number $M > B$ that may be arbitrarily large. The critical advantage is that the total number of model emitters $M$ can now be set before processing, whereas the total number of actual emitters $B$ remains unknown. With this formulation, we infer the values of $b^{1:M}$ during processing simultaneously with the other parameters of interest. In this way, we can actively recruit (that is, $b^m = 1$) or discard (that is, $b^m = 0$) light emitters consistently avoiding underfitting/overfitting. After image processing, our analysis recovers the total number of imaged emitters by the sum $B = \sum_{m=1}^{M} b^m$ and the positions of the emitters $X^m(t)$, $Y^m(t)$, $Z^m(t)$ by the estimated positions of the model emitters with $b^m = 1$. However, a side effect of introducing $M$ is that the results of our analysis may depend on the particular value chosen. To relax this dependence, we use a specialized nonparametric prior on $b^m$ that we describe in detail in Supplementary Information. This prior specifically allows for image processing at the formal limit $M \to \infty$.

Our overall formulation also includes additional parameters (for example, background photon flux and fluorophore brightness) that may or may not be of immediate interest. To provide a flexible computational scheme that works around both unknown types (that is, parametric and nonparametric) and also allows for future extensions, we adopt a Bayesian approach in which we prescribe prior probability distributions on every unknown parameter beyond just the indicators $b^m$. These priors, combined with the preceding formulation, lead to the posterior probability distribution $p(\theta | w_{1:N}^{1:P})$, where $\theta$ gathers every unknown, on which our results rely. We describe the full posterior distribution and its evaluation in Supplementary Information.

After generating samples from the posterior probability distribution $p(\theta | w_{1:N}^{1:P})$ (see Supplementary Note 10 for details), numerous, often thousands of, instances of $\theta$ are acquired. Each $\theta$ contains values for every variable of interest. Subsequently, by aggregating all samples corresponding to each variable, their respective 95% CIs can be computed as the range between the 2.5th and 97.5th percentiles.

## TrackMate, u-track and Tracking Performance Measure
Besides its widespread use, ongoing maintenance and updates and being built upon leading methods in Chenouard et al.[34], we opt for TrackMate[30] specifically because it combines various localization and linking methods and multiple thresholding options.

To generate tracks for comparison in Results, we first export simulated movies as TIFF files and import them in Fiji[60] v1.54b for analysis with TrackMate v7.9.2. As part of implementing TrackMate, the Laplacian of Gaussian detector with subpixel localization and the linear assignment problem mathematical framework[29] are used in spot detection. Spots are then filtered based on quality, contrast, sum intensity and radius (based on the actual PSF size used in data simulation). For the linear assignment problem tracker, we allow gap closing and tune based on diffusion coefficients, the parameters for maximum (interframe) distance, maximum frame gap and the number of spots in tracks to find the best tracks. No extra feature penalties are added. All aforementioned parameters are tuned to minimize tracking errors.

u-track[29] is used for additional comparison, following the same tuning process described earlier. As an advantage to u-track, we provide u-track with the background emission (termed 'absolute background' in the manual), which we instead learn from the simulated data using BNP-Track.

The benchmarks in Supplementary Tables 2 and 3 were created using the Tracking Performance Measure[34] plugin in Icy[46] 2.4.3.0. We exported the TrackMate track, the BNP-Track MAP estimates and ground truth tracks as XML files to generate these benchmarks. All tracks were imported into Icy's TrackManager using the 'Import from TrackMate' feature, and the Tracking Performance Measure plugin was started using the 'add Track Processor' option. The only required input for this plugin is the 'maximum distance between detections' (gate value), for which we used the following three values: 2, 5 and 10 pixels.

Manual connection of track segments was also performed in TrackManager. Initially, we established links between track segments by dragging the last piece of one segment to the first piece of the next segment in the left panel ('Track View') of TrackManager. To ensure recognition of these manual connections by the Tracking Performance Measure plugin, we also selected 'Edit' and 'Fuse All track segments'.

## Image acquisition
**Experimental timelapse images.** Fluorescence timelapse images of U-2 OS (HTB-96, ATCC) cells injected with chemically labeled firefly luciferase mRNAs were acquired simultaneously on two cameras. This cell line was genotyped for authentication and subjected to biweekly mycoplasma contamination checks. This U-2 OS cell line is not in the list of known misidentified cell lines. Cell culture and handling of U-2 OS cells before injections were performed as previously described[61]. Firefly luciferase mRNAs were in vitro transcribed, capped and polyadenylated, and a variable number of Cy3 dyes were nonspecifically added to the poly(A) tail using Click chemistry[39]. Cells were injected with a solution of Cy3-labeled mRNAs and Cascade Blue-labeled 10-kDa dextran (Invitrogen, D1976) using a Femtojet pump and Injectman NI2 micromanipulator (Eppendorf) at 20 hPa for 0.1 s with 20 hPa of compensation pressure. Cells that were successfully injected were identified by the presence of a fluorescent dextran and were imaged 30 min after injection. The cells were continuously illuminated with a 532-nm laser in HILO mode, and Cy3 fluorescence was collected using a ×60/1.49-NA oil objective. Images were captured simultaneously on

two Andor X-10 EMCCD cameras using a 50:50 beamsplitter with a 100-ms exposure time.

**Synthetic timelapse images.** We acquire validation and benchmarking data through standard computer simulations. We start from ground truth as specified in the captions of Figs. 3 and 4 and Supplementary Figs. 2 to 9 and then added noise with values that we estimated from the experimental timelapse images according to Supplementary Note 4.

#### Reporting summary

Further information on research design is available in the Nature Portfolio Reporting Summary linked to this article.

## Data availability

All data discussed in this manuscript are provided as Supplementary Data. Source data are provided with this paper.

## Code availability

All algorithms are implemented in MATLAB R2022b[62], tested in R2024a. Code can be accessed at the GitHub page for the S.P. laboratory at https://github.com/LabPresse/BNP-Track (ref. 63). Figure-generating scripts are in MATLAB or Julia[64] v1.10.3 using Makie[65] v0.20.9. These scripts are available upon request.

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

## Acknowledgements

We acknowledge NIH NIGMS R01GM130745 (to S.P.) for supporting early efforts in nonparametrics and tracking, R01GM134426 (to S.P.) for supporting single-photon efforts, R35GM148237 (to S.P.) entitled 'Toward high spatiotemporal resolution models of single molecules for in cellulo applications' combining both prior R01s and NSF 2310610 (to S.P.) for providing support to build the lab's expertise in competing tracking tools (in the context of bacterial tracking). We also acknowledge R01GM122803 and R35GM131922 (to N.G.W.) for enabling experimental data acquisition as well as the NSF MRI-ID grant DBI-0959823 (to N.G.W.) for seeding the Single Molecule Analysis in Real-Time (SMART) Center, whose Single Particle Tracker TIRFM equipment was used for acquiring experimental tracking data with support from J. D. Hoff. NIH T-32-GM007315 partially supported A.P.J.

## Author contributions

I.S. and L.W.Q.X. prepared the manuscript. I.S., L.W.Q.X. and Z.K. contributed to the analysis methods, computational implementation and software development. A.P.J. and N.G.W. provided the experimental data and provided feedback on the manuscript. I.S. and S.P. conceived the research, and S.P. supervised the entire project.

## Competing interests

S.P., I.S. and L.W.Q.X. are co-inventors on a patent application that incorporates the methods outlined in this manuscript. S.P. is a cofounder at Saguaro Solutions. The other authors declare no competing interests.

## Additional information

**Extended data** is available for this paper at https://doi.org/10.1038/s41592-024-02349-9.

**Correspondence and requests for materials** should be addressed to Steve Pressé.

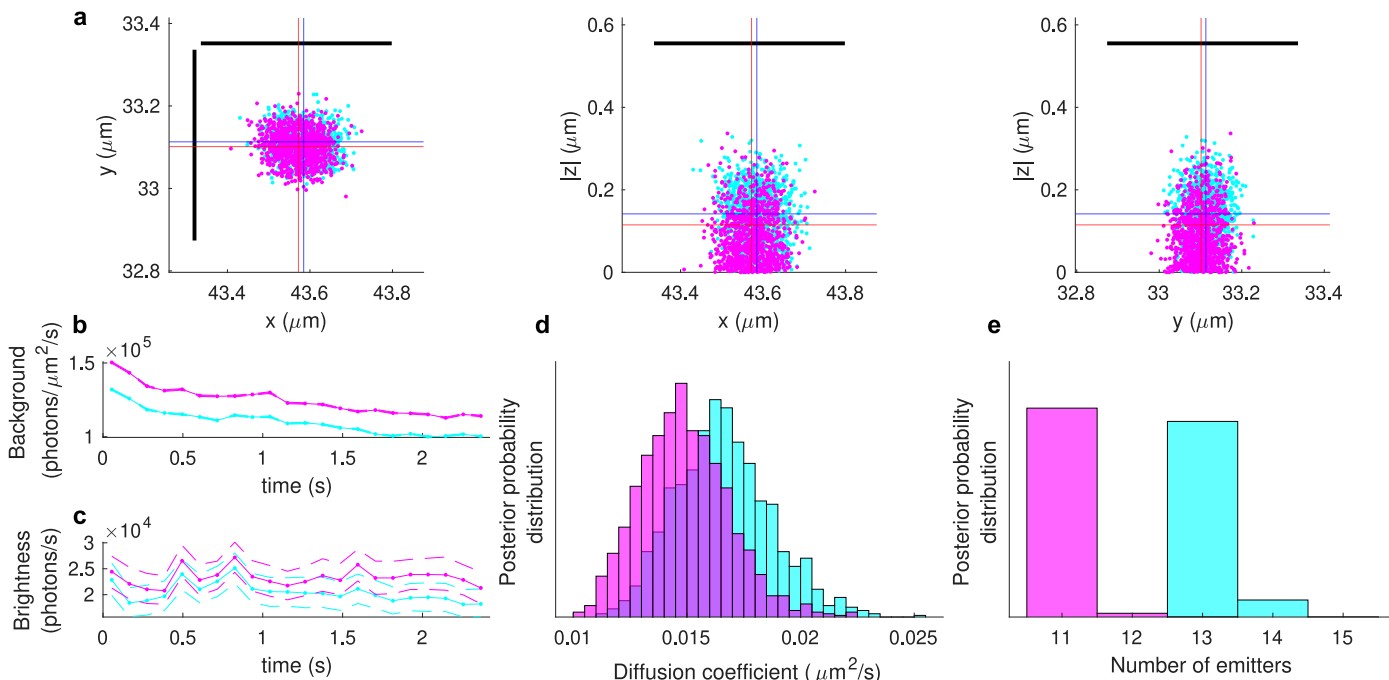

**Extended Data Fig. 1 | Testing BNP-Track's performance for ROI-2.** ROI-2 is shown in Fig. 2, and the color scheme here is the same as in Fig. 2 (cyan for camera A and magenta for camera B). **a**, Localization estimates in the lateral and axial directions at a selected frame of a selected emitter. Dots indicate individual positions sampled from the joint posterior distribution (as detailed in Methods), and blue and red crosses indicate average values for cyan and magenta, respectively. The black line segments mark the diffraction limit in the lateral direction. **b** and **c**, Estimated background photon fluxes (**b**) and emitter brightnesses (**c**) for both cameras throughout imaging. Dotted lines represent median estimates, and dashed lines map the 1%-99% credible interval. **d** and **e**, The posterior distributions of the diffusion coefficient (**d**) and number of emitters (**e**) for both cameras.

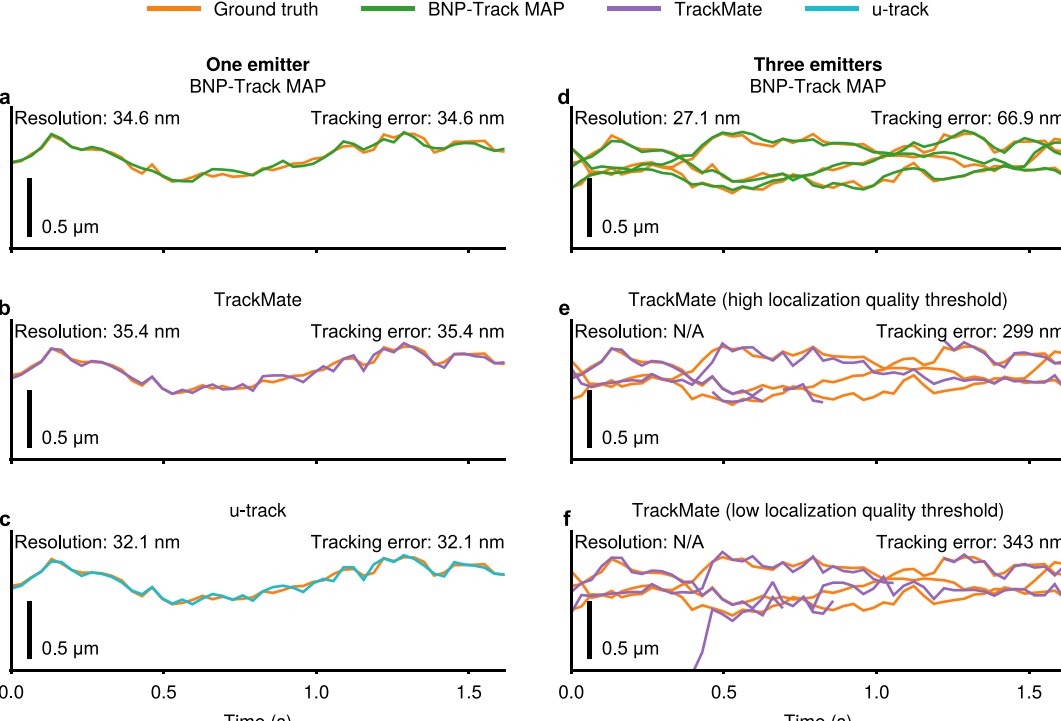

**Extended Data Fig. 2 | Tracking performance comparison among BNP-Track, TrackMate, and u-track.** Same layout as Fig. 3 but for the x coordinate. **a**, BNP-Track's MAP estimate compared to the one-emitter ground truth. **b**, TrackMate's estimate compared to the one-emitter ground truth. Sets A and B are equivalent in this case. **c**, u-track's estimate compared to the one-emitter ground truth. **d**, BNP-Track MAP estimates compared to the three-emitter ground truth. **e**,**f**, TrackMate estimates compared to the three-emitter ground truth with high localization quality threshold (**e**) and low localization quality threshold (**f**). TrackMate estimates compared to the three-emitter ground truth, respectively. In **d**–**f**, the top track of ground truth is the same as the ground truth in **a**–**c**. The boxed regions in **e** and **f** highlight areas where TrackMate performs relatively poorly. See Supplementary Data 5 and 6. Localization resolution is not applicable (N/A) to TrackMate estimates due to missing track segments.

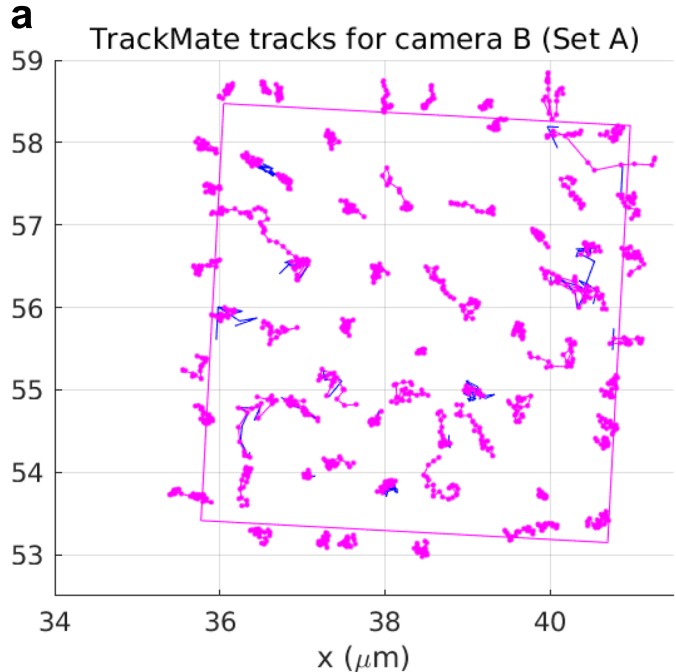

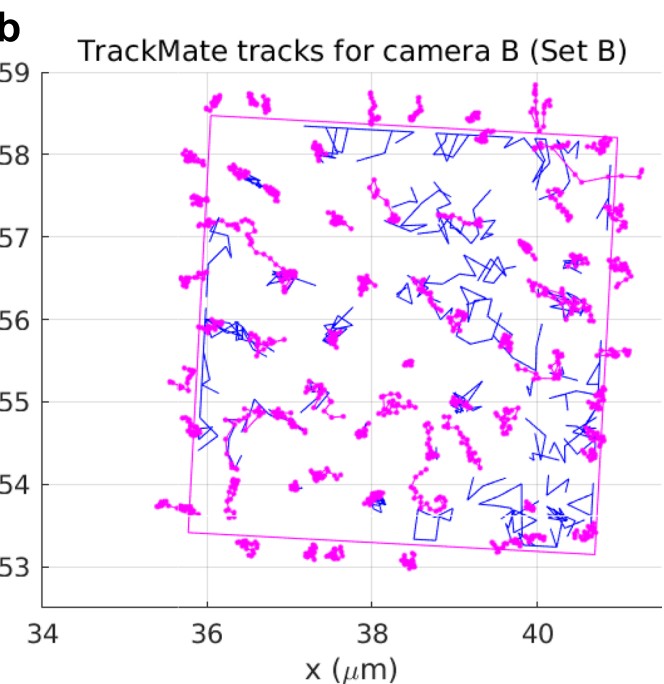

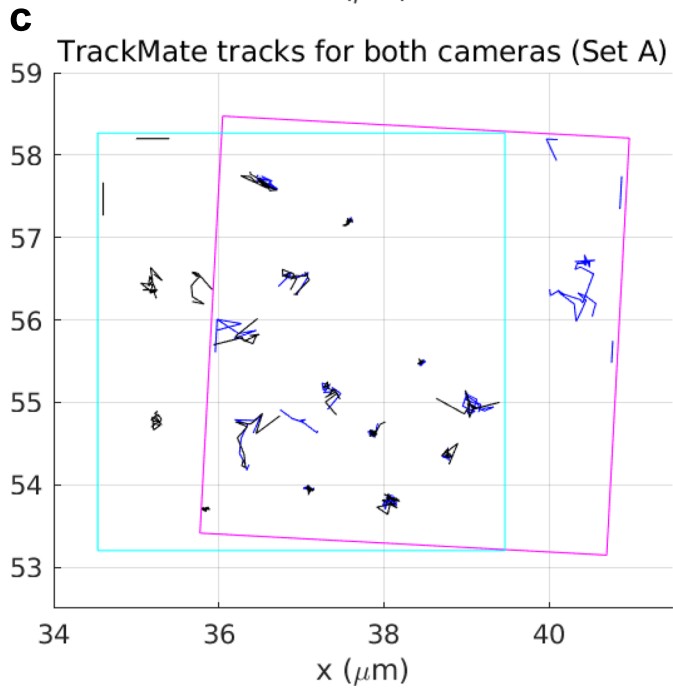

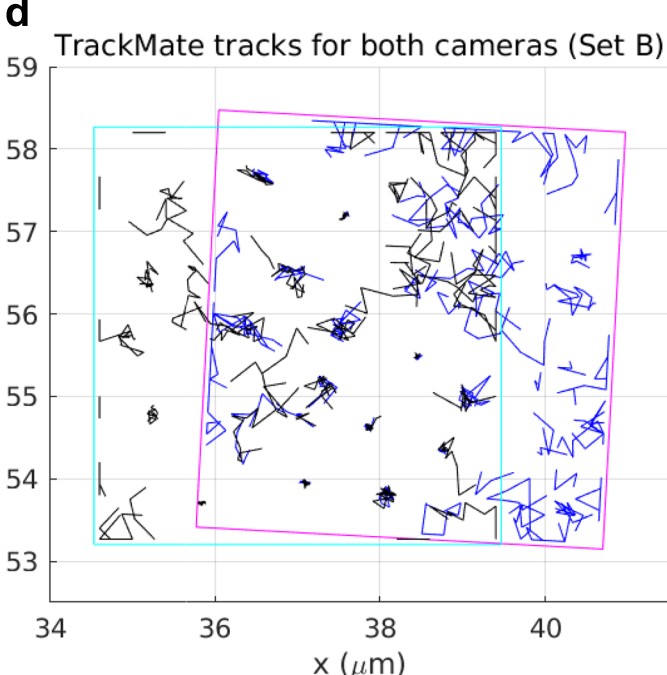

**Extended Data Fig. 3 | BNP-Track's performance in increasingly crowded environments. a**, A comparison between BNP-Track's result and TrackMate's result with a high localization quality threshold (Set A) for the data from camera B. TrackMate tracks are in blue. **b**, A comparison between BNP-Track's result and

TrackMate's result with a low localization quality threshold (Set B) for the data from camera B. TrackMate tracks are in blue. **c** Comparison between TrackMate tracks (Set A) from both cameras with the same image registration as Fig. 5. **d**, Comparison between TrackMate tracks (Set B) from both cameras.

# Reporting Summary

## Statistics

For all statistical analyses, confirm that the following items are present in the figure legend, table legend, main text, or Methods section.

| n/a | Confirmed | |
|---|---|---|
| ☐ | ☒ | The exact sample size (*n*) for each experimental group/condition, given as a discrete number and unit of measurement |
| ☐ | ☒ | A statement on whether measurements were taken from distinct samples or whether the same sample was measured repeatedly |
| ☒ | ☐ | The statistical test(s) used AND whether they are one- or two-sided *Only common tests should be described solely by name; describe more complex techniques in the Methods section.* |
| ☒ | ☐ | A description of all covariates tested |
| ☒ | ☐ | A description of any assumptions or corrections, such as tests of normality and adjustment for multiple comparisons |
| ☐ | ☒ | A full description of the statistical parameters including central tendency (e.g. means) or other basic estimates (e.g. regression coefficient) AND variation (e.g. standard deviation) or associated estimates of uncertainty (e.g. confidence intervals) |
| ☒ | ☐ | For null hypothesis testing, the test statistic (e.g. *F*, *t*, *r*) with confidence intervals, effect sizes, degrees of freedom and *P* value noted *Give P values as exact values whenever suitable.* |
| ☐ | ☒ | For Bayesian analysis, information on the choice of priors and Markov chain Monte Carlo settings |
| ☒ | ☐ | For hierarchical and complex designs, identification of the appropriate level for tests and full reporting of outcomes |
| ☒ | ☐ | Estimates of effect sizes (e.g. Cohen's *d*, Pearson's *r*), indicating how they were calculated |

*Our web collection on statistics for biologists contains articles on many of the points above.*

## Software and code

Policy information about availability of computer code

| Data collection | Synthetic data for this study was generated using a custom MATLAB2022a code, available at https://github.com/LabPresse/BNP-Tracks. Both synthetic and experimental data discussed in the manuscript are available in the Supplementary Information. |
|---|---|
| Data analysis | Available at https://github.com/LabPresse/BNP-Track, TrackMate v7.9.2, u-track 2.3.0, Fiji 1.54b, Icy 2.4.3.0 |

For manuscripts utilizing custom algorithms or software that are central to the research but not yet described in published literature, software must be made available to editors and reviewers. We strongly encourage code deposition in a community repository (e.g. GitHub). See the Nature Portfolio guidelines for submitting code & software for further information.

## Data

Policy information about availability of data

All manuscripts must include a data availability statement. This statement should provide the following information, where applicable:
- Accession codes, unique identifiers, or web links for publicly available datasets
- A description of any restrictions on data availability
- For clinical datasets or third party data, please ensure that the statement adheres to our policy

All data are provided in the Supplementary Videos.

## Human research participants

Policy information about studies involving human research participants and Sex and Gender in Research.

| | |
|---|---|
| Reporting on sex and gender | N/A |
| Population characteristics | N/A |
| Recruitment | N/A |
| Ethics oversight | N/A |

Note that full information on the approval of the study protocol must also be provided in the manuscript.

# Field-specific reporting

Please select the one below that is the best fit for your research. If you are not sure, read the appropriate sections before making your selection.

☒ Life sciences    ☐ Behavioural & social sciences    ☐ Ecological, evolutionary & environmental sciences

For a reference copy of the document with all sections, see nature.com/documents/nr-reporting-summary-flat.pdf

# Life sciences study design

All studies must disclose on these points even when the disclosure is negative.

| | |
|---|---|
| Sample size | No sample-size calculations were needed in this study. In the manuscript, we use synthetic data to verify our analysis method's robustness under various parameters. |
| Data exclusions | No data were excluded. |
| Replication | We guarantee reproducible analysis by provision of our custom data analysis code. Refer to the works cited herein for information on experimental reproducibility.<br>S. Pitchiaya, V. Krishnan, T. C. Custer, and N. G. Walter. Dissecting non-coding rna mechanisms in cellulo by single-molecule high-resolution localization and counting. Methods, 63(2):188, 2013. |
| Randomization | In the manuscript, we use synthetic data to verify our analysis method's robustness under various parameters. Therefore, no randomization is needed. |
| Blinding | Though we rely on data from studies where blinding is relevant, the data we generate result from computer simulations, and therefor blinding does not apply. |

# Reporting for specific materials, systems and methods

We require information from authors about some types of materials, experimental systems and methods used in many studies. Here, indicate whether each material, system or method listed is relevant to your study. If you are not sure if a list item applies to your research, read the appropriate section before selecting a response.

## Materials & experimental systems

| n/a | Involved in the study |
|---|---|
| ☒ | ☐ Antibodies |
| ☐ | ☒ Eukaryotic cell lines |
| ☒ | ☐ Palaeontology and archaeology |
| ☒ | ☐ Animals and other organisms |
| ☒ | ☐ Clinical data |
| ☒ | ☐ Dual use research of concern |

## Methods

| n/a | Involved in the study |
|---|---|
| ☒ | ☐ ChIP-seq |
| ☒ | ☐ Flow cytometry |
| ☒ | ☐ MRI-based neuroimaging |

# Eukaryotic cell lines

Policy information about cell lines and Sex and Gender in Research

| | |
|---|---|
| Cell line source(s) | U-2 OS cells were from ATCC (HTB-96). This is a cell line with epithelial morphology that was derived in 1964 from a moderately differentiated sarcoma of the tibia of a 15-year-old, White, female osteosarcoma patient. |
| Authentication | U2-OS cells were genotyped. |
| Mycoplasma contamination | U-2 OS cells were subjected to biweekly mycoplasma contamination check, and they tested negative. |
| Commonly misidentified lines (See ICLAC register) | No commonly misidentified cell lines were used in this study. |

