## [Peer Review File · Nature Methods]

Peer Review Information

Manuscript Title: BNP-Track: A framework for superresolved tracking

Corresponding author name(s): Steve Presse

Editorial Notes: None

Reviewer Comments & Decisions:

Decision Letter, initial peer review version:

Dear Steve,

Please let me begin by apologizing for the extreme delay on this review process. One of your refs ended up not being able to review, and we had to wait on the second reviewer.

Your Article, "BNP-Track: A framework for superresolved tracking", has now been seen by two reviewers. As you will see from their comments below, although the reviewers find your work of considerable potential interest, they have raised a number of concerns. We are interested in the possibility of publishing your paper in Nature Methods, but would like to consider your response to these concerns before we reach a final decision on publication.

We therefore invite you to revise your manuscript to address these concerns. We found the comments constructive and reasonable on the whole, and we ask you to focus on (1) better explaining/justifying the underlying math, (2) exploring how parameters affect performance, (3) determining upper limits of density for which the method works, and (4) showing densely labeled examples with clear particle crossings.

[Redacted]

We hope to receive your revised paper within three months. If you cannot send it within this time, please let us know. In this event, we will still be happy to reconsider your paper at a later date so long as nothing similar has been accepted for publication at Nature Methods or published elsewhere.

OPEN SCIENCE REQUIREMENTS

REPORTING SUMMARY AND EDITORIAL POLICY CHECKLISTS

Please note that these forms are dynamic ‘smart pdfs’ and must therefore be downloaded and completed in Adobe Reader. We will then flatten them for ease of use by the reviewers. If you would like to reference the guidance text as you complete the template, please access these flattened versions at <http://www.nature.com/authors/policies/availability.html>.

DATA AVAILABILITY

All novel DNA and RNA sequencing data, protein sequences, genetic polymorphisms, linked genotype and phenotype data, gene expression data, macromolecular structures, and proteomics data must be deposited in a publicly accessible database, and accession codes and associated hyperlinks must be provided in the “Data Availability” section.

Please include a “Data availability” subsection in the Online Methods. This section should inform readers about the availability of the data used to support the conclusions of your study, including accession codes to public repositories, references to source data that may be published alongside the paper, unique identifiers such as URLs to data repository entries, or data set DOIs, and any other statement about data availability. At a minimum, you should include the following statement: “The data that

support the findings of this study are available from the corresponding author upon request”, describing which data is available upon request and mentioning any restrictions on availability. If DOIs are provided, please include these in the Reference list (authors, title, publisher (repository name), identifier, year). For more guidance on how to write this section please see: <http://www.nature.com/authors/policies/data/data-availability-statements-data-citations.pdf>

CODE AVAILABILITY

Please include a “Code Availability” subsection in the Online Methods which details how your custom code is made available. Only in rare cases (where code is not central to the main conclusions of the paper) is the statement “available upon request” allowed (and reasons should be specified).

For more information on our code sharing policy and requirements, please see: <https://www.nature.com/nature-research/editorial-policies/reporting-standards#availability-of-computer-code>

MATERIALS AVAILABILITY

OK TO DELETE SECTION IF SUPPLEMENTARY PROTOCOL NOT NEEDED

SUPPLEMENTARY PROTOCOL

To help facilitate reproducibility and uptake of your method, we ask you to prepare a step-by-step Supplementary Protocol for the method described in this paper. We encourage authors to share their step-by-step experimental protocols on a protocol sharing platform of their choice and report the protocol DOI in the reference list. Nature Portfolio 's Protocol Exchange is a free-to-use and open

resource for protocols; protocols deposited in Protocol Exchange are citable and can be linked from the published article. More details can found at www.nature.com/protocolexchange/about.

ORCID

Nature Methods is committed to improving transparency in authorship. As part of our efforts in this direction, we are now requesting that all authors identified as ‘corresponding author’ on published papers create and link their Open Researcher and Contributor Identifier (ORCID) with their account on the Manuscript Tracking System (MTS), prior to acceptance. This applies to primary research papers only. ORCID helps the scientific community achieve unambiguous attribution of all scholarly contributions. You can create and link your ORCID from the home page of the MTS by clicking on ‘Modify my Springer Nature account’. For more information please visit www.springernature.com/orcid.

Sincerely,
Rita

Rita Strack, Ph.D.
Senior Editor
Nature Methods

Reviewers' Comments:

Reviewer #1:

Remarks to the Author:

The paper from the pressse lab and collaborators entitled “BNP-Track: A framework for super-resolved tracking ” introduces a new global Bayesian approach including all steps from identification to tracking of molecules in crowded conditions. The paper is interesting but i was not able to fully understand in what regime it migh bring something instrumental in experiment analysis. I have multiple question to attempts to clarify that.

Major

As density increases, the function to be optimised on the tracking components is no longer the maximum likelihood but rather the partition function in which we sum over all possible paths the particles may have taken. Knowing that the real path is not accessible. Can the author specify the limit density of the approach? The notion of the real path within their framework? I would recommend benchmarking their approach with Micha and Lenka's approach

(<https://www.pnas.org/doi/10.1073/pnas.0910994107>). Yet, I am aware they did not care about localisation

Often in the example shown by the authors, the density of the images is in fact not that high. The images are crowded for sure, with some overlap in the PSFs. Yet we may ask, would an iterative localisation method plus tracking be as efficient as this method (while not be elegant)? Similarly, the density does not seem high enough to induce numerous trajectory crossings.

Something troubles me a bit with the comparison with Trackmate and it relates to the quantity that will be analysed from data. Does single particle identity matter? At high crowding the identity will be lost and it is why partition function need to be optimized. If the quantity of interest is a physical parameters associated to the environment then identity may not matter. Conversely, if identity is what matters, crossing events will lose identity. Then, what is to be tested is not necessarily the percentage of the misclassified tracks but the price in features estimation from the particles.

While there is a lot of work in this paper, it seems that the space of parameters tested is a bit too restricted to gather a full view of its use. Especially in what set of parameters would another approach provide similar results in feature estimation and the current method outperform in a meaningful way the estimated parameters.

From the paper it is not clear whether I should be looking for this approach when data is of poor quality, very noisy with a lot of uncertainties, or just when the data are very crowded.

When addressing crowded data, it would be meaningful to see the transition from rare crossing events, crossing nearly close to every other frame and very high dense and close to what correlation microscopy should handle. It would give an insight of the capacity of the posterior to handle high degree of uncertainties

Another element that would be interesting to explore a little bit more is anomalous detection. Usually in large scale recordings there spots appearing with various size and emission intensity that are not related to the experiments. How does the algorithm handle false positive detections or strong deviation from priors on expected light intensity of the fluorophores?

The approach is computationally intensive, have the authors considered a form of amortisation to accelerate convergence and sampling?

The supplementary lacks a proper discussion of important parameters estimation outside the BNP procedure. At the end of the pipeline how relevant parameters are better estimated using this approach rather than usual ones that are done in multiple steps

Minor

Could the author comment on computing time vs crowding ?

Figure 5 it is a bit difficult to see the overlay. Can you zoom in regions where something meaningful happens.

Figure 2 images seem of low quality (because of the complexity of the data) is that what the method is intended to address. Deformed PSF, big spots and small spots

Reviewer #3:

Remarks to the Author:

A. Summary of the key results

The authors developed a framework to extend superresolution to simultaneous multiple emitter tracking based on nonparametric Bayesian inference. Most well-known superresolution techniques such as STORM, PALM, and PAINT are limited to static samples, that is, at least during the measurements samples to be measured should be dynamically unchanged. Another super-resolution technique based on mathematical modelling is compressed sensing that assumes sparsity in some data space, but again most standard compressed sensing approaches are also limited to static samples. The authors demonstrated their posterior distribution quantifies the uncertainty in emitter numbers and their linkages over time course in synthetic data and in experimental data, reflecting experimental noises, camera artefacts, out-of-focus motion etc. They utilized two different cameras to ensure the reconstructed tracking trajectories are consistent for different sample viewpoints, which validates implicitly their framework. I believe that the manuscript may have a broad impact to the community of Nature Methods. However, there exist several parts to be revised in order to make them clearer.

B. Originality and significance: if not novel, please include reference

The authors' originality is to extend superresolution technique to simultaneous multiple emitter tracking based on nonparametric Bayesian inference, which was difficult and limited for most superresolution techniques. The authors introduced a strategy to monitor the consistency between the reconstructed tracked results in terms of different cameras to monitor the same physical systems in which no one can access the ground truth tracking trajectories in principle. Their framework was validated to some extent via the consistency, regarded as a necessary condition to support their modeling. The authors

demonstrated that even in crowded environments their BNP-Track provides more consistent tracked results than the other SPT methods.

C.Data & methodology: validity of approach, quality of data, quality of presentation

A controversial issue in such mathematical reconstruction schemes is the plausibility and the validity of the mathematical assumptions chosen in each of actual applications. Compared to assuming sparsity of some data space (e.g., physical space, difference space, Fourier space) utilized in compressed sensing for static samples, this study's framework requires more assumptions on optics/motion model/... although the mathematical framework the authors developed may be general once the assumptions and the motion model are validated. The authors showed that BNP-Track provided consistent tracking results across two cameras for an experimental dataset, which can be regarded as a necessary condition to validate the model they assumed.

Page 4: although the criterion Eq. (2) was taken from Ref. [14], the performance comparison is dependent on the choice of the gate value ϵ . The authors used five pixels (approximately 665nm) but there exists no reason of this choice. The authors should explain the reason and argue how the evaluation performance depends on the choice of ϵ in the comparison of different methods such as BNP-track and other SPT methods.

Page 5: the authors wrote "Using the metric defined above for fig. 2e, we report a tracking error (pairing distance averaged over the number of frames) of 73nm in the lateral direction. Consequently, BNP-Track's average error from the underlying ground truth is one half of the tracking error,...". I guess that this ground truth is not ground truth tracking trajectories but it is not so clearly written what means by the underlying ground truth. The reconstructed tracking trajectories made from two images taken by two different cameras should coincide to each other in principle, yielding the metric Eq. (2) being zero. I guess that this is the underlying ground truth which may be one of the unique ideas of the manuscript, if this is the case, but not well written. Likewise in Page 11 the authors also stated "a tracking error of 68.2nm compared to the ground truth," which should also be clarified, i.e., what means by the ground truth.

Page 6-7 on comparing BNP-Track to other SPT methods:

This is one of the key components of this paper to claim the superiority in the performance of BNP-Track to other SPT methods. The authors state manual tuning of the parameters of the other SPT tools to have them best match the ground truth emitter numbers and locations. Manual tuning depends on a person to tune the parameters, and has no guarantee to reproduce the data objectively by definition. Why did not the authors employ a Bayesian optimization to more objectively tune these parameters? I think the other SPT methods may be hard to differentiate emitters close to each other shorter than diffraction limit, and are not designed to reproduce the emitter numbers. Thus, I wonder if to tune the parameters

so as to best match the ground truth emitter numbers and locations below the limit causes troubles in a fair comparison. The authors should clarify this is not the case.

D. Appropriate use of statistics and treatment of uncertainties

Page 9: It is unclear how the authors computed 95% confidence intervals. Also, in the caption of Fig. 4, they used not "confidence intervals" but "credible intervals", which seems to be inconsistent in the usage of words. The authors should clarify how they computed 95% confidence intervals and if they used some mathematical modelling to estimate confidence intervals, they should explain the validity to assume the model for confidence intervals.

E. Conclusions: robustness, validity, reliability

The authors state in Page 13 that "if we have reason to believe that a specific motion model is warranted that may not be accommodated by Gaussian transition probabilities, we may also incorporate this change into our framework." It would not be desired to require a user to provide a reason to warrant the choice of a specific motion model before measurements. For example, the assumption of diffusion constant D being spatially constant entirely across the field of view may be too simplified. Isn't it impossible to employ a more general diffusion model and to naturally make the measurement speak for themselves which diffusion model is most plausible for a given sample to measure? I believe that, in order to warrant feasibility of this framework, not just imposing a set of models to represent optics/motion model/... by a user, a model selection built-in framework is desired to autonomously extract the underlying plausible model from a sample. The authors should address such possibility and, if possible, provide some demonstrations.

Page 13. the authors claim that "we have demonstrated that BNP-Track yields accurate tracking results consistent across two cameras for an experimental dataset with an unknown emitter motion model, despite assuming Brownian motion. This may suggest that BNP-Track remains robust under other motion models." I cannot understand the logical connection between the two sentences. I agree that to result in consistent tracking results across the two cameras is regarded as a necessary condition to support the free Brownian motion (although the further model such as position-dependent diffusion model may further improve the consistency), but why the present consistent results can suggest that BNP-Track remains robust under other unexamined motion models? I think there exists a logical jump. The authors provided implicit evidence that free Brownian motion model presented consistent BNP tracking results (to some extent) across different two cameras, but did not provide evidence (or at least thorough discussions) that further modification of motion model has no improvement in extracting the tracking trajectories (i.e., model selection problem). The authors should address the model selection problem especially in the motion model and how their framework can/cannot spontaneously choose the most appropriate model.

F. Suggested improvements: experiments, data for possible revision

Some of the suggestions were already written and the following is additional questions and comments.

In page 20 in SI: The authors modeled $U_{\text{back}}(x,y,t) = C(t)$ as a uniform-in-space flux, i.e., irradiance profile does provide position-independent constant intensity across the field of view as for the background, which seems not to be trivial. How did the authors validate this assumption?

In page 22 in SI: The authors modeled the temporal discretization by mid-point rule instead of integrations. Some discussions on the appropriateness of this approximation are also required.

In page 23 in SI: The authors modeled the emitter motion to be purely free Brownian, in which diffusion constants are not dependent on positions and time. I expect that the author's framework can extend to model selection problem instead of imposing a single model for representing the underlying motions. If tracked regimes are not so large in physical space, difference in diffusion constants may not impact the consistency of the reconstructed tracked results. The authors should address how to generalize their framework to model selection.

Minor points:

Page 4 just after Eq. (2): θ_n should be ψ_n .

Page 7: Although the authors wrote "even as these fall below the diffraction limit in frames 2 to 13 and 34 to 47 (see fig. A.2a)", it is better to use the time because the corresponding figures are depicted along the time unit.

Page 10: Fig. 5, it is very difficult to find how tracked trajectories in out-of-focus direction (z) are correctly matched between two cameras. The authors had better redraw either of the tracked results using cameras A and B as transparent color.

Page 11: the authors state "we set a relative high localization quality (Set A) threshold at 5" but what is the unit of the 5?

G. References: appropriate credit to previous work?

I think the reference provides appropriate credit to previous works.

H. Clarity and context: lucidity of abstract/summary, appropriateness of abstract, introduction and conclusions

Abstract/summary/introduction and conclusions are considered to be clearly well written although some revisions are required as stated above.

We thank the referees for their thoughtful comments and the Editor for their synthesis of the feedback. We have addressed all comments resulting in five added figures and multiple clarifying comments across the text and SI. We have numbered the comments in blue, e.g., RXY which stands for Referee X-Comment Y, and referred to them by comment number in the main text to help orient the referees.

Point-by-point response to reviewers:

Reviewer #1:

Remarks to the Author:

R1C1 The paper from the presse lab and collaborators entitled "BNP-Track: A framework for super-resolved tracking" introduces a new global Bayesian approach including all steps from identification to tracking of molecules in crowded conditions. The paper is interesting but i was not able to fully understand in what regime it might bring something instrumental in experiment analysis. I have multiple question to attempts to clarify that.

We thank this reviewer for the helpful comments. Please find point-by-point responses below.

Major

R1C2 As density increases, the function to be optimised on the tracking components is no longer the maximum likelihood but rather the partition function in which we sum over all possible paths the particles may have taken. Knowing that the real path is not accessible. Can the author specify the limit density of the approach?

We directly address this important point on page 13 through simulations that involve comparisons against ground truth. Briefly, we have incorporated SI fig. A.7, wherein we maintain the field of view's area and progressively increase the number of emitters. The results numerically showcase that our framework remains effective until there are no more than 10 emitters within an approximately $6 \mu\text{m}^2$ region. Additionally, we introduce a dimensionless metric termed "crowdedness" to quantify emitter number density. For instance, having 10 emitters in around $6 \mu\text{m}^2$ corresponds to a crowdedness of approximately 0.4, given the parameter we employed in data generation.

R1C3 The notion of the real path within their framework? I would recommend benchmarking their approach with micha and Lenka's approach (<https://www.pnas.org/doi/10.1073/pnas.0910994107>). Yet, i am aware they did not care about localisation.

We thank the reviewer for this comment and have highlighted, on page 3, where we cite the recommended paper as one example of an SPT tool that performs emitter number determination, emitter localization, and linking in separate steps. In addition, as for the notion of real path, we have clarified on page 7 that we now perform extensive tests using synthetic datasets with known ground truth tracks and parameters.

Regarding a direct and fair benchmark comparison with the message-passing belief propagation (BP) algorithm by Chertkov et al., we contend that its implementation presents challenges. As noted by the reviewer, the BP algorithm lacks localization capabilities, making its performance directly reliant on the quality of localization. The first option involves supplying emitter locations obtained from a particle localization tool to the BP algorithm. However, as discussed in our paper, conducting localization independently on each frame raises difficulties when emitter PSFs significantly overlap, placing the BP algorithm at a notable disadvantage.

The alternative approach entails employing a method, let's say method X, that does not separate localization and linking, and subsequently providing the BP algorithm with only the emitter positions. Nevertheless, there are two crucial considerations: i) BNP-Track stands out as the sole known candidate for method X, and ii) it would be more appropriate to compare method X directly to BNP-Track under such a scenario.

R1C4 Often in the example shown by the authors, the density of the images is in fact not that high. The images are crowded for sure, with some overlap in the PSFs. Yet we may ask, would an iterative localisation method plus tracking be as efficient as this method (while not be elegant)? Similarly, the density does not seem high enough to induce numerous trajectory crossings.

We thank the reviewer for this comment. Firstly, we have now clarified in the caption of fig. 5 that we achieve a high local emitter number density by bringing two emitter tracks into closer proximity until the mean displacement between them reaches zero. Additionally, as in our response to R1C2, we have conducted and presented supplementary tests (Supplementary Movies 24–26) that delve into the "crowdedness" of the system. Moreover, for each SI figure involving three emitters (figs A.3, A.8–A10), we have incorporated a plot illustrating pairwise distances between emitter positions over time. This plot demonstrates that there are consistently multiple frames where two emitters exist in closer proximity than the nominal diffraction limit.

In response to the reviewer's query regarding the iterative localization method plus tracking, indeed certain techniques, as exemplified by Chenouard et al.'s approach employing multiple hypothesis tracking, purport to execute iterative tracking. Nevertheless, we could not find a readily available tool founded upon these iterative methodologies. The nearest alternatives we identified are methods such as TrackMate. At any rate, a paragraph has been incorporated into the Discussion section, noting that conventional localization-then-linking methods may be favored in scenarios where bright emitters exhibit substantial separation.

R1C5 Something troubles me a bit with the comparison with Trackmate as it relates to the quantity that will be analysed from data. Does single particle identity matter? At high crowding the identity will be lost and it is why partition function need to be optimized. If the quantity of interest is a physical parameters associated to the environment then identity may not matter. Conversely, if identity is what matters, crossing events will lose identity. Then, what is to be tested is not necessarily the percentage of the misclassified tracks but the price in features estimation from the particles.

We are thankful to the reviewer for this comment, and in response, we now emphasize, on page 8, that emitters may lose their identities and become indistinguishable when their PSFs significantly overlap.

Moreover, regarding the reviewer's query about the influence of mislinks in TrackMate on feature estimation, we have introduced two additional paragraphs on page 10, within section "Comparison with TrackMate". These paragraphs exemplify how such mislinks can alter the outcomes of subsequent analyses, using the diffusion coefficient as an illustrative example. In addition, diffusion coefficient

estimates derived from TrackMate tracks and MSD have been included in fig. 4, A.3–A.10. In nearly all instances, BNP-Track demonstrates better accuracy and consistency in estimating diffusion coefficients.

Despite this supplementary examination, we have opted to maintain our original metric, as it aligns with the established standard in the cited literature (Chenouard et al., Nat. Methods, 2014).

R1C6 While there is a lot of work in this paper, it seems that the space of parameters tested is a bit too restricted to gather a full view of its use. Especially in what set of parameters would another approach provide similar results in feature estimation and the current method outperform in a meaningful way the estimated parameters.

We thank the reviewer for this comment. In response, we have incorporated figs A.5–A.7 into the SI to show the parameter regimes wherein our framework proves valid. These supplementary tests are comprehensively addressed in the main text, specifically on page 12. Notably, we have conducted thorough robustness tests encompassing emitter brightness, background flux, and particle number density. The inclusion of the first two is particularly beneficial, given their direct impact on a dataset's signal-to-noise ratio, while particle number density significantly influences the occurrence of PSF overlaps and trajectory intersections.

Moreover, we have included figs. A.9–A.11 to illustrate the applicability of BNP-Track to systems with motion models different from the assumed 3D normal diffusion, as discussed in the main text on page 13 within a newly introduced section titled "Robustness tests for motion models". Specifically, these tests consider scenarios that will be of interest to this referee in which we investigate diffusion coefficients different amongst emitters or otherwise exhibiting spatial or temporal dependencies.

R1C7 From the paper it is not clear whether I should be looking for this approach when data is of poor quality, very noisy with a lot of uncertainties, or just when the data are very crowded.

This is an excellent point, as mentioned in our response to R1C4, we have added a paragraph in blue to Discussion on page 17. Briefly, therein we discuss the fact that different existing approaches apply to limited regimes. Yet these regimes naturally assume limited out-of-focus background, brighter fluorophores, and lower density. In realistic images, we cannot control the fact that some molecules will appear at the focus (and thus appear high SNR) while others will be much dimmer. Any kind of real data, which is naturally heterogeneous, will therefore benefit from this type of method. These points are also made clearer thanks to the robustness analyses added in response to this reviewer's previous comments.

R1C8 When addressing crowded data, it would be meaningful to see the transition from rare crossing events, crossing nearly close to every other frame and very high density and close to what correlation microscopy should handle. It would give an insight of the capacity of the posterior to handle high degree of uncertainties.

We appreciate this insightful comment. In response, we have enhanced the clarity of the caption in fig. 5. The tracks of two emitters are depicted, gradually converging from a mean displacement over two times the nominal diffraction limit (indicating no crossing) to zero (representing numerous crossings).

R1C9 Another element that would be interesting to explore a little bit more is anomalous detection. Usually in large scale recordings there are spots appearing with various sizes and emission intensities that are not related to the experiments. How does the algorithm handle false positive detections or strong deviation from priors on expected light intensity of the fluorophores?

We have now added a paragraph in the Discussion highlighting how our framework currently attributes variations in spot size and emission intensity to (i) emitters moving into and out of the focal plane, (ii) overlapping emitters, and (iii) motion blur. In large part, given the nature of the data, we now explain that it is difficult to ascertain the origin of spots not coming from molecules as these would be reported with large uncertainty either way as would be positions of molecules out of focus. Put differently, given the data at hand, no method can currently identify the nature of far out of focus spots. To help address this, we briefly discuss scanning light sheet in the discussion helping place different planes in focus.

R1C10 The approach is computationally intensive, have the authors considered a form of amortisation to accelerate convergence and sampling?

We thank the reviewer for this suggestion. As a result of this comment, we have now expanded the Discussion section on page 16 regarding computational cost amortization from parallelism. Briefly, as the average number of photons incident on each pixel is independent of the others (see eq. 6), Employing parallelism can, theoretically, speed-up our algorithm by a factor equal to the smaller of the two numbers: the total number of pixels and the number of processor cores.

R1C11 The supplementary lacks a proper discussion of important parameters estimation outside the BNP procedure. At the end of the pipeline how relevant parameters are better estimated using this approach rather than usual ones that are done in multiple steps

We hope that our response to this referee's previous comments, in particular R1C5 and R1C6, have now helped address this concern.

Minor

R1C12 Could the author comment on computing time vs crowding?

With fixed field of view, the computational cost scales roughly linearly with the number of particles. This is now highlighted on page 16 in Discussion.

R1C13 Figure 5 it is a bit difficult to see the overlay. Can you zoom in regions where something meaningful happens.

We thank the reviewer for this comment. Fig. 5 is now modified by zooming into average frames.

R1C14 Figure 2 images seems of low quality (because of the complexity of the data) is that what the method is intended to address. Deformed PSF, big spots and small spots

We thank the reviewer for this comment. We addressed this comment by adding statement to the main text (page 5) to the effect that fig. 2 "appears complex as it reflects the nature of the data to which BNP-Track applies".

Reviewer #3:

Remarks to the Author:

A. Summary of the key results

R3C1 The authors developed a framework to extend superresolution to simultaneous multiple emitter tracking based on nonparametric Bayesian inference. Most well-known superresolution techniques such as STORM, PALM, and PAINT are limited to static samples, that is, at least during the measurements samples to be measured should be dynamically unchanged. Another super-resolution technique based on mathematical modelling is compressed sensing that assumes sparsity in some data space, but again most

standard compressed sensing approaches are also limited to static samples. The authors demonstrated their posterior distribution quantifies the uncertainty in emitter numbers and their linkages over time course in synthetic data and in experimental data, reflecting experimental noises, camera artefacts, out-of-focus motion etc. They utilized two different cameras to ensure the reconstructed tracking trajectories are consistent for different sample viewpoints, which validates implicitly their framework. I believe that the manuscript may have a broad impact to the community of Nature Methods. However, there exist several parts to be revised in order to make them clearer.

We thank this reviewer for the comments, please see our point-by-point response below.

B. Originality and significance: if not novel, please include reference

The authors' originality is to extend superresolution technique to simultaneous multiple emitter tracking based on nonparametric Bayesian inference, which was difficult and limited for most superresolution techniques. The authors introduced a strategy to monitor the consistency between the reconstructed tracked results in terms of different cameras to monitor the same physical systems in which no one can access the ground truth tracking trajectories in principle. Their framework was validated to some extent via the consistency, regarded as a necessary condition to support their modeling. The authors demonstrated that even in crowded environments their BNP-Track provides more consistent tracked results than the other SPT methods.

C. Data & methodology: validity of approach, quality of data, quality of presentation

R3C2 A controversial issue in such mathematical reconstruction schemes is the plausibility and the validity of the mathematical assumptions chosen in each of actual applications. Compared to assuming sparsity of some data space (e.g., physical space, difference space, Fourier space) utilized in compressed sensing for static samples, this study's framework requires more assumptions on optics/motion model/...

This is indeed an important pedagogical point. While not necessarily a question by the referee, we do highlight for this referee (though not in the text) that because we make our assumptions explicit, it does appear at face-value that we have more assumptions. However, these controlled assumptions are necessary in avoiding the ad hoc thresholds and other adjustable parameters of other methods which are more difficult to control and physically motivate.

although the mathematical framework the authors developed may be general once the assumptions and the motion model are validated. The authors showed that BNP-Track provided consistent tracking results across two cameras for an experimental dataset, which can be regarded as a necessary condition to validate the model they assumed.

R3C3 Page 4: although the criterion Eq. (2) was taken from Ref. [14], the performance comparison is dependent on the choice of the gate value ϵ . The authors used five pixels (approximately 665nm) but there exists no reason of this choice. The authors should explain the reason and argue how the evaluation performance depends on the choice of ϵ in the comparison of different methods such as BNP-track and other SPT methods.

We thank the reviewer for this comment. We have added a paragraph on page 8 explaining the origin of our choice of the gate value ϵ . Briefly, as the gate value represents the "maximum allowed distance for two detections to be pair", we set it such that "well-separated" detections, or those exceeding twice the nominal diffraction limit are never paired. Since the nominal diffraction limit is about 280 nm, we chose ϵ equals five pixels.

We have now also expanded tables B.2 and B.3 with varying values for ϵ demonstrating that changing ϵ does not appreciably alter the performance comparison presented in our manuscript.

R3C4 Page 5: the authors wrote "Using the metric defined above for fig. 2e, we report a tracking error (pairing distance averaged over the number of frames) of 73nm in the lateral direction. Consequently, BNP-Track's average error from the underlying ground truth is one half of the tracking error,..." I guess that this ground truth is not ground truth tracking trajectories but it is not so clearly written what means by the underlying ground truth. The reconstructed tracking trajectories made from two images taken by two different cameras should coincide to each other in principle, yielding the metric Eq. (2) being zero. I guess that this is the underlying ground truth which may be one of the unique ideas of the manuscript, if this is the case, but not well written. Likewise in Page 11 the authors also stated "a tracking error of 68.2nm compared to the ground truth," which should also be clarified, i.e., what means by the ground truth.

We thank the reviewer for highlighting this confusion. We have now added a paragraph on page 5 explaining what we mean by error between both detectors and have eliminated the phrase "ground truth" in describing the experimental data.

R3C5 Page 6-7 on comparing BNP-Track to other SPT methods: This is one of the key components of this paper to claim the superiority in the performance of BNP-Track to other SPT methods. The authors state manual tuning of the parameters of the other SPT tools to have them best match the ground truth emitter numbers and locations. Manual tuning depends on a person to tune the parameters, and has no guarantee to reproduce the data objectively by definition. Why did not the authors employ a Bayesian optimization to more objectively tune these parameters? I think the other SPT methods may be hard to differentiate emitters close to each other shorter than diffraction limit, and are not designed to reproduce the emitter numbers. Thus, I wonder if to tune the parameters so as to best match the ground truth emitter numbers and locations below the limit causes troubles in a fair comparison. The authors should clarify this is not the case.

We thank the reviewer for this comment. Indeed, we now clarify on page 7 that instead of optimizing all tracks to their theoretical optima, all tools (including BNP-Track) are given enough and equal amount of time to fine-tune their hyper-parameters (developing a fully rigorous way to improve competing tools may be beyond the scope of the current work and would not coincide with how people use u-track or TrackMate).

On the same page, we also clarify that we manually tune other SPT tools' parameters and fuse track segments in order to give them the strongest possible advantage and, in doing so, avoid a straw-man argument.

D. Appropriate use of statistics and treatment of uncertainties

R3C6 Page 9: It is unclear how the authors computed 95% confidence intervals. Also, in the caption of Fig. 4, they used not "confidence intervals" but "credible intervals", which seems to be inconsistent in the usage of words. The authors should clarify how they computed 95% confidence intervals and if they used some mathematical modelling to estimate confidence intervals, they should explain the validity to assume the model for confidence intervals.

We thank the reviewer for this comment. In response, we have added a paragraph on page 19 within the "Model inference" section. In essence, numerous samples (often exceeding thousands) are drawn for each variable of interest from their associated posterior. Subsequently, the 95% credible interval is

determined by the range spanning from the 2.5th to the 97.5th percentiles of the respective variable's samples. As credible intervals represent posterior probability distributions' breadth, they are informed by all uncertainties captured by the likelihood, detailed in SI section H, including photon shot noise, detector noise, finiteness of data, and pixelization.

Also, we have changed the "confidence interval" in fig. 4's caption to "credible interval". We apologize for this confusion.

E. Conclusions: robustness, validity, reliability

R3C7 The authors state in Page 13 that "if we have reason to believe that a specific motion model is warranted that may not be accommodated by Gaussian transition probabilities, we may also incorporate this change into our framework." It would not be desired to require a user to provide a reason to warrant the choice of a specific motion model before measurements. For example, the assumption of diffusion constant D being spatially constant entirely across the field of view may be too simplified. Isn't it impossible to employ a more general diffusion model and to naturally make the measurement speak for themselves which diffusion model is most plausible for a given sample to measure? I believe that, in order to warrant feasibility of this framework, not just imposing a set of models to represent optics/motion model/. . . by a user, a model selection built-in framework is desired to autonomously extract the underlying plausible model from a sample. The authors should address such possibility and, if possible, provide some demonstrations.

We thank the referee for this critical point addressed in a few different ways. First, and most simply, we note in the Discussion that the nature of the data itself is sufficiently limited such that learning the motion model (in addition to everything else learned including trajectories and numbers of particles) may not be possible (technically it would give rise to high uncertainty over the candidate motion models).

Instead, akin to our reply regarding R1C6, we have added a section on page 13 along with SI figs. A.8–A.10. These illustrate that, while the actual tracks themselves are not drawn from a normal diffusion model, we can still consistently track and learn particle numbers using a Gaussian transition probability (amounting to a normal diffusion model). Briefly, this happens because we have enough photon budget to localize particles though we may not have enough to discriminate one motion model from another.

R3C8 Page 13. the authors claim that "we have demonstrated that BNP-Track yields accurate tracking results consistent across two cameras for an experimental dataset with an unknown emitter motion model, despite assuming Brownian motion. This may suggest that BNP-Track remains robust under other motion models." I cannot understand the logical connection between the two sentences. I agree that to result in consistent tracking results across the two cameras is regarded as a necessary condition to support the free Brownian motion (although the further model such as position-dependent diffusion model may further improve the consistency), but why the present consistent results can suggest that BNP-Track remains robust under other unexamined motion models? I think there exists a logical jump.

Indeed this is a point we have now clarified more thoroughly from the demonstration on Fig. 2 of page 5 where there is no reason to believe *a priori* that the motion model dictating the data that we subsequently analyze is normal diffusive.

R3C9 The authors provided implicit evidence that free Brownian motion model presented consistent BNP tracking results (to some extent) across different two cameras, but did not provide evidence (or at least thorough discussions) that further modification of motion model has no improvement in extracting the tracking trajectories (i.e., model selection problem). The authors should address the model selection

problem especially in the motion model and how their framework can/cannot spontaneously choose the most appropriate model.

In response to this excellent comment, we have generated necessary SI Figs. A.9–A.11. For these datasets, we allow diffusion coefficient to vary across space, time, or across particles. The results shown in these aforementioned figures numerically demonstrate that our framework is not sensitive to motion models.

F. Suggested improvements: experiments, data for possible revision

Some of the suggestions were already written and the following is additional questions and comments.

R3C10 In page 20 in SI: The authors modeled $U_{\text{back}}(x,y,t) = C(t)$ as a uniform-in-space flux, i.e., irradiance profile does provide position-independent constant intensity across the field of view as for the background, which seems not to be trivial. How did the authors validate this assumption?

We thank the referee for this comment. Indeed this is a subtle point brought up differently by the previous referee (R1C9).

In large part, given the nature of the data, we now explain in the Discussion that it is difficult to ascertain the origin of spots (attributed to background or other features) not coming from out-of-focus molecules as these would be reported with large uncertainty. Put differently, given the limited nature of the data at hand, no method can currently discriminate what background features are truly background versus far out of focus molecules. To help address this, we briefly discuss scanning light sheet in the Discussion as well helping place different planes in focus and break this problem's intrinsic degeneracy.

R3C11 In page 22 in SI: The authors modeled the temporal discretization by mid-point rule instead of integrations. Some discussions on the appropriateness of this approximation are also required.

This is a great point. We have now added a discussion of this point on page 24 in SI to clarify why an analytical evaluation of the time integral is unattainable. Furthermore, on SI page 26 in section G.5, we now highlight the potential for enhancing the precision of our approximations through the application of the composite trapezoidal rule. In the limit where the number of subintervals approaches infinity, the approximation error converged to zero with a quadratic rate. We provide validation of these numerical approximations in SI Fig. A.4, where we evaluate the impact of varying the numbers of subintervals (sampled positions) on an emitter moving at a rate of $1 \mu\text{m}^2/\text{s}$ with a 30 ms camera exposure.

R3C12 In page 23 in SI: The authors modeled the emitter motion to be purely free Brownian, in which diffusion constants are not dependent on positions and time. I expect that the author's framework can extend to model selection problem instead of imposing a single model for representing the underlying motions. If tracked regimes are not so large in physical space, difference in diffusion constants may not impact the consistency of the reconstructed tracked results. The authors should address how to generalize their framework to model selection.

We have now discussed how to integrate different motion models such as hop-diffusion and have added the model to our SI (G.4) though sampling from such a model is difficult. However, just as our response to R3C7, we have also added a comment to the effect that learning both motion model in addition to particle number and trajectories may be too demanding given the nature of the data.

Minor points:

R3C13 Page 4 just after Eq. (2): θ_n should be ψ_n .

Please see edit on page 4 now made.

R3C14 Page 7: Although the authors wrote “even as these fall below the diffraction limit in frames 2 to 13 and 34 to 47 (see fig. A.2a)”, it is better to use the time because the corresponding figures are depicted along the time unit.

We have now uniformized notation, now on page 10, as well as other similar places.

R3C15 Page 10: Fig. 5, it is very difficult to find how tracked trajectories in out-of-focus direction (z) are correctly matched between two cameras. The authors had better redraw either of the tracked results using cameras A and B as transparent color.

We thank the referee for this comment. We have now modified Fig. 5 such that the tracks are now plotted with lower opacity colors.

R3C16 Page 11: the authors state “we set a relative high localization quality (Set A) threshold at 5” but what is the unit of the 5?

We have added “pixels” as unit on page 15. To make this number more intuitive, we also convert this number to “nanometers” in the main text.

G. References: appropriate credit to previous work?

I think the reference provides appropriate credit to previous works.

H. Clarity and context: lucidity of abstract/summary, appropriateness of abstract, introduction and conclusions

Abstract/summary/introduction and conclusions are considered to be clearly well written although some revisions are required as stated above.

Decision Letter, second revision:

Dear Steve,

Thank you for submitting your revised manuscript "BNP-Track: A framework for superresolved tracking" (NMETH-A52505B). It has now been seen by the original referees and their comments are below. The reviewers find that the paper has improved in revision, and therefore we'll be happy in principle to publish it in Nature Methods, pending minor revisions to comply with our editorial and formatting guidelines.

TRANSPARENT PEER REVIEW

Please note: we allow redactions to authors' rebuttal and reviewer comments in the interest of confidentiality. If you are concerned about the release of confidential data, please let us know specifically what information you would like to have removed. Please note that we cannot incorporate redactions for any other reasons. Reviewer names will be published in the peer review files if the reviewer signed the comments to authors, or if reviewers explicitly agree to release their name. For more information, please refer to our FAQ page.

ORCID

Sincerely,
Rita

Rita Strack, Ph.D.
Senior Editor

Nature Methods

Reviewer #3 (Remarks to the Author):

The authors responded to all comments I made. Especially, one of the most important revisions was to demonstrate the applicability of BNP-Track for other motion models for superresolution techniques to simultaneous multiple emitter tracking in time domain as free as possible from falling into a unique motion model which has inevitably a potential bias. The following is my comments on the revised version of the manuscript.

Page 8. 6th line: There exists typo.

"we will this SPT tool to output two sets of". The verb is missing.

Page 8. 19th line: The authors wrote "As we will show, even providing competing methods significant advantages, BNP-Track still exceeds the resolution of existing tools and yields reduced error rates (percentage of wrong links)." This is a not adequate description. For example, in Table B.3 for gate value 2, The RMSE for BNP-Track MAP is 0.518 but that for TrackMate A is 0.424, which would tell that TrackMate A has a higher resolution than BNP-Track MAP, whilst the number of missed detections in TrackMate A is much more (62) than that (8) in BNP-Track MAP. The authors should revise the statement more precisely.

Page 14: The authors wrote "BNP-Track's estimate of the diffusion coefficient should be compared to an average ground truth value." This seems to be a too strong statement, and the authors should modify the statement. The ground truth would be masked by the weighted average in general, because the ground truth is space-dependent, time-dependent, or different emitters, which cannot be characterized by average ground truth value in full. I expect that BNP-Track is capable of predicting, for example, where, for what time the underlying diffusion constant changes. This should however be very complicated because the CI are subject not only to diffusion constant but also to space and time where the diffusion constant changes. Thus, I interpret that the authors decided not to going to such details and used in average ground truth value for the sake of brevity. The authors had better present such detailed analysis for future work.

These are simply a suggestion for expressions and future work. The present contribution is already sufficiently novel and interesting for publication in Nature Methods.

Author Rebuttal, third revision:

We thank the referees for their thoughtful comments and the Editor for their synthesis of the feedback.

Point-by-point response to reviewers:

Reviewer #3:

Remarks to the Author:

The authors responded to all comments I made. Especially, one of the most important revisions was to demonstrate the applicability of BNP-Track for other motion models for superresolution techniques to simultaneous multiple emitter tracking in time domain as free as possible from falling into a unique motion model which has inevitably a potential bias. The following is my comments on the revised version of the manuscript.

R3C1 Page 8. 6th line: There exists typo. "we will this SPT tool to output two sets of". The verb is missing.

We thank the reviewer for this comment. In the revised manuscript on page 5, it is now "we output two sets of tracks for this SPT tool from data".

R3C2 Page 8. 19th line: The authors wrote "As we will show, even providing competing methods significant advantages, BNP-Track still exceeds the resolution of existing tools and yields reduced error rates (percentage of wrong links)." This is a not adequate description. For example, in Table B.3 for gate value 2, The RMSE for BNP-Track MAP is 0.518 but that for TrackMate A is 0.424, which would tell that TrackMate A has a higher resolution than BNP-Track MAP, whilst the number of missed detections in TrackMate A is much more (62) than that (8) in BNP-Track MAP. The authors should revise the statement more precisely.

We thank the reviewer for this comment. On page 8, we have changed the statement to "BNP-Track still exceeds the resolution of existing tools in most cases".

R3C3 Page 14: The authors wrote "BNP-Track's estimate of the diffusion coefficient should be compared to an average ground truth value." This seems to be a too strong statement, and the authors should modify the statement. The ground truth would be masked by the weighted average in general, because the ground truth is space-dependent, time-dependent, or different emitters, which cannot be characterized by average ground truth value in full. I expect that BNP-Track is capable of predicting, for example, where, for what time the underlying diffusion constant changes. This should however be very complicated because the CI are subject not only to diffusion constant but also to space and time where the diffusion constant changes. Thus, I interpret that the authors decided not to going to such details and used in average ground truth value for the sake of brevity. The authors had better present such detailed analysis for future work.

We thank the reviewer for this comment. On page 18 of the revised Supplementary Discussion, we now clarify that if a method assumes one constant diffusion coefficient, the best estimate it can produce would be the average diffusion coefficient. In the same paragraph, we now also clarify that further

extensions of BNP-Track, which incorporate changes in diffusion coefficient, despite being possible, are out of the scope of this paper.

R3C4 These are simply a suggestion for expressions and future work. The present contribution is already sufficiently novel and interesting for publication in Nature Methods.

We thank the reviewer for their comments and suggestions!

Final Decision Letter:

Dear Steve,

I am pleased to inform you that your Article, "BNP-Track: A framework for superresolved tracking", has now been accepted for publication in Nature Methods. The received and accepted dates will be May 6, 2023 and June 3, 2024. This note is intended to let you know what to expect from us over the next month or so, and to let you know where to address any further questions.

Over the next few weeks, your paper will be copyedited to ensure that it conforms to Nature Methods style. Once your paper is typeset, you will receive an email with a link to choose the appropriate publishing options for your paper and our Author Services team will be in touch regarding any additional information that may be required. It is extremely important that you let us know now whether you will be difficult to contact over the next month. If this is the case, we ask that you send us the contact information (email, phone and fax) of someone who will be able to check the proofs and deal with any last-minute problems.

Please note that *Nature Methods* is a Transformative Journal (TJ). Authors may publish their research with us through the traditional subscription access route or make their paper immediately open access through payment of an article-processing charge (APC). Authors will not be required to make a final decision about access to their article until it has been accepted. Find out more about Transformative Journals

You may wish to make your media relations office aware of your accepted publication, in case they consider it appropriate to organize some internal or external publicity. Once your paper has been scheduled you will receive an email confirming the publication details. This is normally 3-4 working days in advance of publication. If you need additional notice of the date and time of publication, please let the production team know when you receive the proof of your article to ensure there is

sufficient time to coordinate. Further information on our embargo policies can be found here:
<https://www.nature.com/authors/policies/embargo.html>

If you are active on Twitter/X, please e-mail me your and your coauthors' handles so that we may tag you when the paper is published.

Best regards,
Rita

Rita Strack, Ph.D.
Senior Editor
Nature Methods